# Logarithmic Switching Regret for Online Convex Optimization

**Wenhao Yang** [1 2]  **Yibo Wang** [1 2]  **Yuanyu Wan** [3 4]  **Lijun Zhang** [1 2]

## Abstract

Online convex optimization in non-stationary environments has garnered considerable attention in the literature. Recently, Pasteris et al. (2024) investigate online convex optimization with the optimal *switching regret*, defined as the sum of the static regret over each segment, where the segmentation is an arbitrary partition of the entire time horizon. For general convex functions, their work has established an optimal switching regret bound. However, it remains open whether similar bounds are attainable for other types of convex functions, such as exponentially concave or strongly convex functions. In this paper, we affirmatively answer this question by proposing a novel meta-algorithm, termed IRESET, which is used to aggregate the decisions from a group of experts. The essence of our method lies in running multiple experts over a set of intervals, and then employing a meta-algorithm equipped with second-order bounds to sequentially combine their decisions. We leverage the segment tree structure to analyze the switching regret over the entire time horizon, and offer new insights into utilizing recursive equations over the segment tree. By choosing appropriate expert-algorithms for IRESET, our methods achieve logarithmic switching regret bounds for exponentially concave or strongly convex functions, respectively.

## 1. Introduction

Online convex optimization (OCO) stands as a cornerstone of online learning, providing a fundamental framework for tackling a broad spectrum of sequential prediction and decision-making problems (Hazan, 2016). OCO can be modeled as a repeated game between the learner and the environment, governed by a specific protocol in which, at each round $t \in [T]$, the learner chooses a decision $\mathbf{x}_t$ from a convex set $\mathcal{X} \subseteq \mathbb{R}^d$, where $d$ is the dimensionality. After submitting the decision, the learner receives a loss $f_t(\mathbf{x}_t)$ and observes the gradient feedback, where $f_t \colon \mathcal{X} \mapsto \mathbb{R}$ is a convex function selected by the environment. The learner aims to minimize the cumulative loss over $T$ rounds. To measure the performance, static regret is typically used:

$$\text{REG}_T \triangleq \sum_{t=1}^{T} f_t(\mathbf{x}_t) - \min_{\mathbf{w} \in \mathcal{W}} \sum_{t=1}^{T} f_t(\mathbf{x})$$

which is defined as the difference between the cumulative loss of the online learner and that of the best decision chosen in hindsight. Since the seminal work of Zinkevich (2003), plenty of algorithms have been developed to minimize the static regret, which achieve $O(\frac{d}{\alpha} \log T)$, $O(\frac{1}{\lambda} \log T)$, and $O(\sqrt{T})$ for $\alpha$-exponentially concave (abbr. $\alpha$-exp-concave), $\lambda$-strongly convex, and general convex functions (Shalev-Shwartz, 2011; Hazan, 2016) respectively. However, the environment is continually evolving in many practical applications, which makes static regret less desirable, since it is unrealistic to assume that a single decision would perform satisfactorily throughout the entire time horizon. To address this, recent advances in OCO have introduced stronger performance measures, including dynamic regret (Zinkevich, 2003; Zhang et al., 2018a), adaptive regret (Hazan & Seshadhri, 2007; Daniely et al., 2015) and switching regret (Pasteris et al., 2024). This study focuses on the switching regret. Since adaptive regret and switching regret are closely related, we first review the concept of adaptive regret, and then discuss its connection to switching regret.

### 1.1. Adaptive Regret and Switching Regret

Adaptive regret, first proposed by Hazan & Seshadhri (2007), is defined as the maximum static regret over any interval $I = [r, s] \subseteq [T]$, which measures performance of the online learner with respect to a changing comparator. However, their notion becomes less meaningful for small intervals as it does not account for the length of interval. Daniely et al. (2015) further refine this concept by introduc-

---

[1]State Key Laboratory of Novel Software Technology, Nanjing University, Nanjing, China [2]School of Artificial Intelligence, Nanjing University, Nanjing, China [3]School of Software Technology, Zhejiang University, Ningbo, China [4]Hangzhou High-Tech Zone (Binjiang) Institute of Blockchain and Data Security, Hangzhou, China. Correspondence to: Lijun Zhang <zhanglj@lamda.nju.edu.cn>.

*Proceedings of the 43rd International Conference on Machine Learning*, Seoul, South Korea. PMLR 306, 2026. Copyright 2026 by the author(s).

ing strongly adaptive regret:

$$\text{SA-REG}_T(\tau) \triangleq \max_{t \in [T]} \left[ \sum_{t=s}^{s+\tau-1} f_t(\mathbf{x}_t) - \min_{\mathbf{x} \in \mathcal{X}} \sum_{t=s}^{s+\tau-1} f_t(\mathbf{x}) \right]$$

which is defined as the maximum static regret on $\max_{[s,s+\tau-1] \subseteq [T]}$ over intervals of length $\tau$. In the literature, several online algorithms have been developed to attain $O(\frac{d}{\alpha} \log \tau \log T)$, $O(\frac{1}{\lambda} \log \tau \log T)$, and $O(\sqrt{\tau \log T})$ for $\alpha$-exp-concave, $\lambda$-strongly convex, and general convex functions (Hazan & Seshadhri, 2009; Jun et al., 2017; Zhang et al., 2018b), respectively. It can be observed that the strongly adaptive regret bounds achieved by existing algorithms incur an additional $O(\log T)$ factor, which represents the cost of adapting to the changing environments.

Recently, Pasteris et al. (2024) introduce an alternative interval-based performance measure, termed *switching regret*, which is also referred to as *tracking regret* in the literature under the framework of tracking the best expert (Vovk, 1997; Herbster & Warmuth, 1998; 2001; Bousquet & Warmuth, 2002). We provide the definition of switching regret. A *segmentation* $\mathcal{S}$ is defined as any partition of $[T]$ into segments. The *switching regret* with respect to $\mathcal{S}$ is defined:

$$\text{SW-REG}_T(\mathcal{S}) \triangleq \sum_{\mathcal{I} \in \mathcal{S}} \left[ \sum_{t \in \mathcal{I}} f_t(\mathbf{x}_t) - \min_{\mathbf{x} \in \mathcal{X}} \sum_{t \in \mathcal{I}} f_t(\mathbf{x}) \right]$$

which is the sum of static regret over each segment $\mathcal{I} \in \mathcal{S}$. Notably, strongly adaptive regret can directly imply switching regret. For example, for general convex functions, an existing algorithm achieving an $O(\sqrt{\tau \log T})$ strongly adaptive regret that adapts to all intervals will also imply an $O(\sum_{\mathcal{I} \in \mathcal{S}} \sqrt{|\mathcal{I}| \log T})$ switching regret bound. However, this bound is unsatisfactory because it still exhibits an $O(\sqrt{\log T})$ gap from the minimax switching regret $\Omega(\sum_{\mathcal{I} \in \mathcal{S}} \sqrt{|\mathcal{I}|})$ for general convex functions established by Pasteris et al. (2024, Theorem 2.1).

In the following, we briefly review the meta-expert framework of existing strongly adaptive algorithms, and then provide intuition on how optimal switching regret can be achieved within the meta-expert framework.

**Existing meta-expert framework for strongly adaptive algorithms** Most of existing strongly adaptive algorithms (Hazan & Seshadhri, 2009; Jun et al., 2017; Zhang et al., 2018b; Wan et al., 2022) adopt a meta-expert framework and include three key components:

- an expert-algorithm, which achieves optimal static regret over a given interval;
- a set of intervals, each of which is associated with an expert-algorithm;
- a meta-algorithm, which aggregates the decisions from all active experts.

Notably, adaptive algorithms essentially create and remove experts dynamically, a mechanism termed *sleeping experts* (Freund et al., 1997), which means that the expert can be inactive (or asleep) in certain rounds and thus does not make decisions. Furthermore, the meta-algorithm of adaptive algorithms is required to support sleeping experts, as it needs to combine the outputs from active experts over every interval. Hence, adaptive algorithms typically incur an additional factor, often logarithmic, that scales with the horizon $T$, which is the price for adaptivity to every interval.

Surprisingly, Pasteris et al. (2024) show that this additional factor is *avoidable* when considering switching regret, and establish an optimal $O(\sum_{\mathcal{I} \in \mathcal{S}} \sqrt{|\mathcal{I}|})$ switching regret for general convex functions. In contrast to strongly adaptive algorithms, we highlight two significant differences in their algorithm design: (i) their algorithm initializes all experts at the beginning, and restarts them across different intervals; (ii) their algorithm employs Hedge (Freund & Schapire, 1997) to sequentially (or pairwise) aggregate the decisions of all experts. In this way, their meta-algorithm does not need to support sleeping experts, intuitively circumventing the additional dependency on $T$. Furthermore, the main contribution of their theoretical analysis lies in deriving recursive equations based on a segment tree structure and performing them over the tree.

In this paper, we focus on other types of convex functions, namely $\alpha$-exp-concave or $\lambda$-strongly convex functions. The switching regret bounds of $O(\sum_{\mathcal{I} \in \mathcal{S}} \frac{d}{\alpha} \log |\mathcal{I}| \log T)$ and $O(\sum_{\mathcal{I} \in \mathcal{S}} \frac{1}{\lambda} \log |\mathcal{I}| \log T)$, implied by the $O(\frac{d}{\alpha} \log \tau \log T)$ and $O(\frac{1}{\lambda} \log \tau \log T)$ strongly adaptive regret bounds respectively, still exhibit an $O(\log T)$ gap from the minimax optimal switching regret, as established below.

**Theorem 1.** *The lower bounds for switching regret are* $\Omega(\sum_{\mathcal{I} \in \mathcal{S}} \frac{d}{\alpha} \log |\mathcal{I}|)$ *and* $\Omega(\sum_{\mathcal{I} \in \mathcal{S}} \frac{1}{\lambda} \log |\mathcal{I}|)$ *for $\alpha$-exp-concave and $\lambda$-strongly convex, respectively.*

**Remark.** The above theorem is intuitive. Assuming the segmentation $\mathcal{S}$ is known in advance, we can directly apply an OCO algorithm designed for static regret and restart it at the beginning of each segment, thereby establishing the lower bounds for switching regret.

Since strongly convex functions and exp-concave functions are also of significant interest in the literature, we seek to extend the success of Pasteris et al. (2024) in achieving optimal switching regret for general convex functions (without the typical logarithmic factor in $T$) to these two types of convex functions.

### 1.2. Technical Challenges and Our Contributions

Our investigation confronts notable technical challenges. These challenges are primarily twofold:

- **Algorithm design:** The first challenge resides in the design of the meta-algorithm, as the meta-algorithm of previous work is inadequate for our specific requirement. To elaborate, the Hedge used in Pasteris et al. (2024), or the sleeping coin betting employed in Jun et al. (2017), incurs at least $\Theta(\sqrt{\tau})$ for intervals of length $\tau$, which is tolerable for convex functions but suboptimal for exp-concave and strongly convex functions. Moreover, the Follow-the-leading-history (Hazan & Seshadhri, 2007) delivers an $O(\log s)$ bound for the interval $I = [r, s]$. While this bound is generally tight when considering adaptive regret, it will suffer from an $O(\log T)$ bound for switching regret.
- **Theoretical analysis:** The second challenge stems from the theoretical analysis. While the analysis presented by Pasteris et al. (2024) is effective for general convex functions, it is not readily extendable to exp-concave or strongly convex functions. This is primarily because their analysis is tailored to regret terms with an $O(\sqrt{\tau})$ dependency for an interval of length $\tau$. In contrast, for exp-concave or strongly convex functions, we need to deal with an $O(\log \tau)$ regret. This fundamental difference in scaling renders their analytical techniques unsuitable, necessitating a novel analysis for these functions.

To address the above challenges, this paper introduces new ideas in both algorithm design and theoretical analysis. Our contributions are summarized as below:

- First, we design a novel algorithm, termed IRESET, which can be viewed as an improved version of RESET proposed by Pasteris et al. (2024). Specifically, IRESET creates multiple experts operating on a set of Dense GC intervals (Zhang et al., 2020). Then, inspired by Zhang et al. (2022), we adopt a meta-algorithm possessing a second-order bound, such as Adapt-ML-Prod (Gaillard et al., 2014), and utilize *linearized losses* to evaluate the performance of experts. The final decision is produced by sequentially aggregating the decisions from all active experts.

- Second, we introduce a novel technical analysis tailored specifically for these two types of functions, which involves performing recursive equations over the segment tree. Furthermore, we provide a novel method to convert the tree-based switching regret into a formulation in terms of the segment lengths.

By choosing appropriate expert-algorithms, our proposed methods achieve nearly-optimal switching regret, i.e., $O(\sum_{\mathcal{I} \in \mathcal{S}} \frac{d}{\alpha} \log^2 |\mathcal{I}|)$ and $O(\sum_{\mathcal{I} \in \mathcal{S}} \frac{1}{\lambda} \log^2 |\mathcal{I}|)$ for $\alpha$-exp-concave and $\lambda$-strongly convex functions, respectively. In comparison with the guarantees $O(\sum_{\mathcal{I} \in \mathcal{S}} \frac{d}{\alpha} \log |\mathcal{I}| \log T)$ and $O(\sum_{\mathcal{I} \in \mathcal{S}} \frac{1}{\lambda} \log |\mathcal{I}| \log T)$ implied by strongly adaptive

regret, our results improve the switching regret bounds from $\log T$ to $\log |\mathcal{I}|$, which is nearly optimal relative to the established lower bound in Theorem 1. Finally, we also develop a universal algorithm for minimizing the switching regret, which achieves $O(\sum_{\mathcal{I} \in \mathcal{S}} \sqrt{|\mathcal{I}|})$, $O(\frac{d}{\alpha} \sum_{\mathcal{I} \in \mathcal{S}} \log^2 |\mathcal{I}|)$ and $O(\frac{1}{\lambda} \sum_{\mathcal{I} \in \mathcal{S}} \log^2 |\mathcal{I}|)$ for general convex, $\alpha$-exp-concave and $\lambda$-strongly convex functions simultaneously.

**Organization** The rest is structured as follows. Section 2 presents some preliminaries of this work. Section 3 provides our approach for optimizing switching regret and the refined analysis. Section 5 concludes this work. Proofs are deferred to the appendices.

## 2. Preliminaries

In this section, we introduce the standard assumptions in OCO. Then, we present the setup for the segment tree, which serve as the foundation for our work.

### 2.1. Assumptions

We introduce the following standard assumptions used in the studies of OCO (Hazan, 2016).

**Assumption 1.** *The norm of the gradients of all functions over the domain $\mathcal{X}$ are bounded by $G$ for all $t \in [T]$, i.e., $\max_{\mathbf{x} \in \mathcal{X}} \|\nabla f_t(\mathbf{x})\| \leq G$.*

**Assumption 2.** *The diameter of the domain $\mathcal{X}$ is bounded by $D$, i.e., $\max_{\mathbf{x}, \mathbf{y} \in \mathcal{X}} \|\mathbf{x} - \mathbf{y}\| \leq D$.*

Next, we state the definitions of strongly convexity and exp-concavity (Boyd & Vandenberghe, 2004; Cesa-Bianchi & Lugosi, 2006), and introduce an important property of exp-concave functions.

**Definition 1.** *A function $f : \mathcal{X} \mapsto \mathbb{R}$ is $\lambda$-strongly convex if*

$$f(\mathbf{y}) \geq f(\mathbf{x}) + \langle \nabla f(\mathbf{x}), \mathbf{y} - \mathbf{x} \rangle + \frac{\lambda}{2} \|\mathbf{y} - \mathbf{x}\|^2,$$

*for all $\mathbf{x}, \mathbf{y} \in \mathcal{X}$.*

**Definition 2.** *A function $f : \mathcal{X} \mapsto \mathbb{R}$ is $\alpha$-exp-concave if $\exp(-\alpha f(\cdot))$ is concave over $\mathcal{X}$.*

**Lemma 1** (Lemma 3 of Hazan et al. (2007))**.** *For a function $f : \mathcal{X} \mapsto \mathbb{R}$, where $\mathcal{X}$ has diameter $D$, such that $\forall \mathbf{x} \in \mathcal{X}$, $\|\nabla f(\mathbf{x})\| \leq G$ and $\exp(-\alpha f(\cdot))$ is concave, we have*

$$f(\mathbf{y}) \geq f(\mathbf{x}) + \langle \nabla f(\mathbf{x}), \mathbf{y} - \mathbf{x} \rangle + \frac{\beta}{2} \langle \nabla f(\mathbf{x}), \mathbf{y} - \mathbf{x} \rangle^2,$$

*for all $\mathbf{x}, \mathbf{y} \in \mathcal{X}$, where $\beta = \frac{1}{2} \min\{\frac{1}{4GD}, \alpha\}$.*

### 2.2. Segment Tree: Notations, Definitions and Proposition

Recursion over the segment tree (RESET), proposed by Pasteris et al. (2024), employs a meta-expert framework where

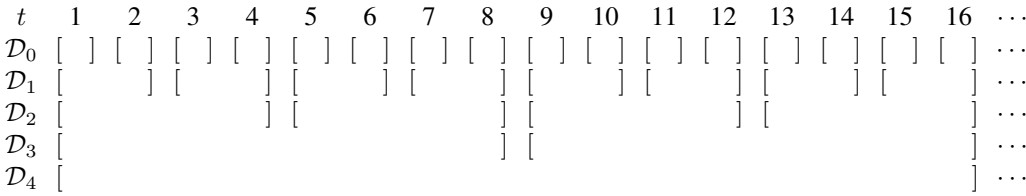

Figure 1. Dense geometric covering (DGC) intervals. In the figure, each interval is denoted by [ ].

multiple experts are maintained and sequentially aggregated by a meta-algorithm. Their algorithm creates $N = O(\log T)$ experts at the beginning, and restarts them at specific time points. This strategy is equivalent to implicitly constructing a set of Dense Geometric Covering (DGC) intervals as illustrated in Figure 1, with each interval corresponding to an expert-algorithm. DGC intervals, introduced by Zhang et al. (2020) for minimizing adaptive and dynamic regret simultaneously, are a dense version of GC intervals (Daniely et al., 2015), which are defined as

$$\mathcal{D} = \bigcup_{k \in \mathbb{N} \cup \{0\}} \mathcal{D}_k \qquad (1)$$

where $\mathcal{D}_k = \{I_k^i = [(i-1) \cdot 2^k + 1, i \cdot 2^k] : i \in \mathbb{N}\}$. The adoption of DGC intervals by Pasteris et al. (2024) is motivated by compatibility with a segment tree structure, which facilitates the analysis of switching regret. In the following, we detail the setup of this segment tree.

Segment tree is a tree data structure commonly used for maintaining information about intervals, enabling efficient operations such as interval queries and summations through recursive equations. This recursive structure proves particularly useful for analyzing the switching regret, which involves aggregating performance over arbitrary partitions of the time horizon. Without loss of generality, we assume $T$ is an integer power of 2. In this case, the set of DGC intervals naturally corresponds to the nodes of a *full, balanced and binary* segment tree, where each node maintains a specific interval.

Denote the segment tree by $\mathcal{B}$ and its root node by $r$. Following the setup of Pasteris et al. (2024), we introduce the following notations and definitions that are utilized throughout our analysis.

**Notations** The following notations are useful.

- $\blacktriangleleft (v)$ is the left-most descendent of $v \in \mathcal{B}$.
- $\blacktriangleright (v)$ is the right-most descendent of $v \in \mathcal{B}$.
- $\uparrow (v)$ is the parent of $v \in \mathcal{B} \setminus r$.
- $h(v)$ is the height of $v \in \mathcal{B}$, and the height of leave node is 0.

**Definitions** Let $\mathcal{S} = \{\mathcal{S}_1, \cdots, \mathcal{S}_k\}$ be an arbitrary segmentation of the time horizon $[1, T]$ into $\Phi = |\mathcal{S}|$ contigu-

ous segments. We define the start and end points of these segments using $\sigma_k$ for $k \in \{1, 2, \cdots, \Phi + 1\}$, such that the $k$-th segment is $[\sigma_k, \sigma_{k+1} - 1]$, where $\Phi = |\mathcal{S}|$, with $\sigma_1 = 1$ and $\sigma_{\Phi+1} - 1 = T$. In the following, we introduce three classifications for vertices in $\mathcal{B}$ relative to the segmentation $\mathcal{S}$:

- **Stationary:** For a vertex $v \in \mathcal{B}$ if there exists $k \in [\Phi]$ with $\sigma_k \leq \blacktriangleleft (v)$ and $\blacktriangleright (v) < \sigma_{k+1}$. Let $\mathcal{H}$ be the set that contains all stationary vertices.
- **Fundamental:** For a vertex $v \in \mathcal{B}$ if it satisfies both (i) $v \in \mathcal{H}$ and (ii) $v = r$ or $\uparrow (v) \notin \mathcal{H}$. Let $\mathcal{F}$ be the set that contains all fundamental vertices.
- **Relevant:** For a vertex $v \in \mathcal{B}$ if it satisfies (i) $v \in \mathcal{F}$ or (ii) it is an ancestor of a fundamental vertex. Let $\mathcal{R}$ be the set that contains all fundamental vertices.

To facilitate understanding, we provide an example to clarify the insight behind the above definitions. Let us consider $T = 8$ and a segmentation $\mathcal{S} = \{[1, 3], [4, 5], [6, 8]\}$ with $\sigma_1 = 1$, $\sigma_2 = 4$, $\sigma_3 = 6$, and $\sigma_4 = 9$. Figure 2 illustrates the vertices corresponding to these three classifications using different colors. First, a stationary vertex represents any node in the segment tree whose maintained interval is entirely contained within a single segment of the given segmentation. Second, a fundamental vertex is a stationary vertex that is "maximal" in the sense that its parent is not stationary (unless it is the root). These vertices essentially form a unique partition of the time horizon using the largest possible stationary intervals from the segment tree. Third, a relevant vertex includes all fundamental vertices and all of their ancestors in the segment tree, up to the root. These vertices form the minimal subtree that connects all fundamental vertices to the root. These definitions are instrumental in the analysis of switching regret.

Based on the segment tree structure, we present the following proposition.

**Proposition 1** (Lemma 4.2 of Pasteris et al. (2024)). *Given any vertex $v \in \mathcal{B}$, we have*

$$\blacktriangleright (v) - \blacktriangleleft (v) + 1 = 2^{h(v)}.$$

*where $\blacktriangleleft (v)$ and $\blacktriangleright (v)$ are the left-most and right-most points of the interval maintained by vertex $v$.*

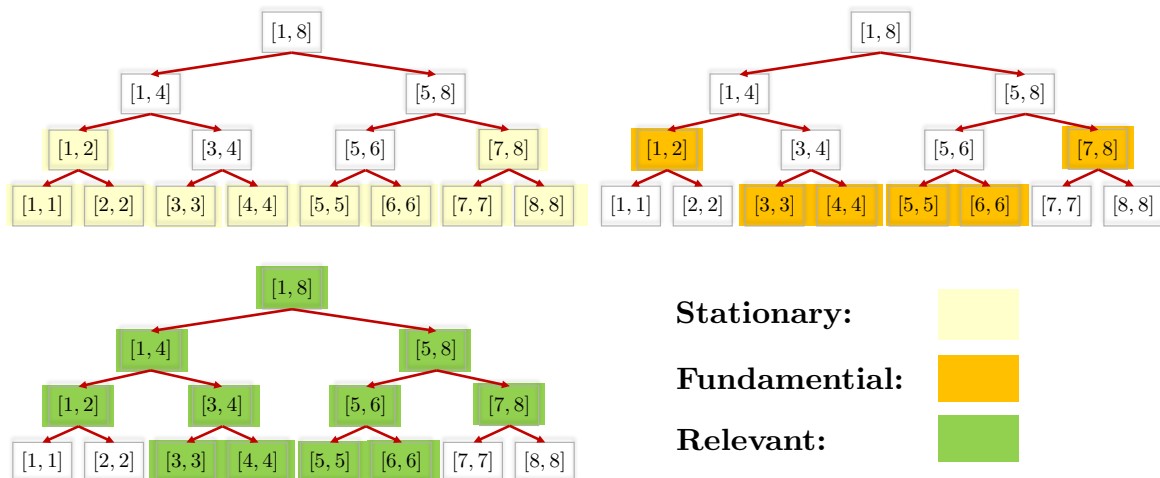

Figure 2. Three classifications for vertices in $\mathcal{B}$ to the segmentation $\mathcal{S}$ for $T = 8$ with segments $\mathcal{S}_1 = [1, 3], \mathcal{S}_2 = [4, 5], \mathcal{S}_3 = [6, 8]$. The vertices corresponding to these three classifications using different colors.

**Remark.** Proposition 1 characterizes the fundamental relationship between the length of the interval maintained by a vertex and its height in the segment tree.

Finally, we would like to clarify that the segment tree structure is employed solely for analytical purposes and not for algorithm execution. Therefore, the time complexity associated with constructing the segment tree does not contribute to the computational complexity of the proposed algorithm.

## 3. Our Approach

In this section, we first introduce our approach, termed IRE-SET. Then, we present our novel analysis. Finally, we propose a universal algorithm for achieving the nearly-optimal switching regret.

### 3.1. An Improved Version of RESET

Following the meta-expert framework for strongly adaptive algorithms, our algorithm also contains three components, including an expert-algorithm, a set of intervals and a meta-algorithm. For the construction of the intervals, we adopt the DGC intervals as defined in (1), which facilitates the analysis of switching regret through the utilization of a segment tree. For the expert-algorithm, we create an expert over each DGC interval by running an instance of OGD (Shalev-Shwartz et al., 2007) or ONS (Hazan et al., 2007) to handle strongly convex or exponentially concave functions, respectively. Then, inspired by Zhang et al. (2022), we choose Adapt-ML-Prod (Gaillard et al., 2014) with linearized loss as the meta-algorithm, which can automatically leverage properties of functions to achieve small meta-regret. We run multiple instances of Adapt-ML-Prod with different param-

eters to sequentially combine the decisions of all experts.

Our Improved version of Recursion over the Segment Tree (IRESET) is summarized in Algorithm 1. In the $t$-th round, we first create an expert $E_i$ for each interval $I = [r, s] \in \mathcal{D}$ that starts from $t$, where $i = \log_2(s - r + 1)$ denotes the index of the expert and its depth in the segment tree, and then introduce two variables $w_{t-1,i}, \widehat{w}_{t-1,i-1}$ to be the updating parameters for Adapt-ML-Prod in Step 6. In Step 7, we maintain a set $\mathcal{A}_t$ consisting of all the active experts, and add $E_i$ into this set. Denote the decision of expert $E_i$ at round $t$ as $\mathbf{x}_{t,i}$. From Step 10 to 18, IRESET sequentially aggregates the decisions from all the experts in ascending order of their indices. Specifically, for the first expert, i.e., the expert with index 0 operating on an interval of length 1, we set $\mathbf{z}_{t,0} = \mathbf{x}_{t,0}$ in Step 12, where $\mathbf{z}_{t,i}$ denotes the intermediate output after aggregating experts up to index $i$. Starting from the second expert, we aggregate their decisions pairwise according to Adapt-ML-Prod. In Step 14, the weights $p_{t,i}$ and $\widehat{p}_{t,i-1}$ for combining two experts are determined by

$$p_{t,i} = \frac{\eta_{t-1,i} w_{t-1,i}}{\eta_{t-1,i} w_{t-1,i} + \widehat{\eta}_{t-1,i-1} \widehat{w}_{t-1,i-1}},$$

$$\widehat{p}_{t,i-1} = \frac{\widehat{\eta}_{t-1,i-1} \widehat{w}_{t-1,i-1}}{\eta_{t-1,i} w_{t-1,i} + \widehat{\eta}_{t-1,i-1} \widehat{w}_{t-1,i-1}} \quad (2)$$

where $p_{t,i} + \widehat{p}_{t,i-1} = 1$, and $\eta_{t,i}, \widehat{\eta}_{t-1,i-1}$ are defined as

$$\eta_{t,i} = \min\left\{\frac{1}{2}, \sqrt{\frac{\ln 2}{1 + L_{t-1,i}}}\right\},$$

$$\widehat{\eta}_{t,i-1} = \min\left\{\frac{1}{2}, \sqrt{\frac{\ln 2}{1 + \widehat{L}_{t-1,i-1}}}\right\}.$$

Then, we output the intermediate decision $\mathbf{z}_{t,i}$ in Step 15. In Step 16, we observe the following linearized losses for the

**Algorithm 1** An Improved version of RESET (IRESET)

1: Initialize the active set: $\mathcal{A}_0 = \emptyset$, and the number of experts: $K = 0$
2: **for** $t = 1$ to $T$ **do**
3:     $\mathcal{A}_t = \mathcal{A}_{t-1}$
4:     **for all** $I = [r, s] \in \mathcal{D}$ that starts from $t$ **do**
5:         Obtain $i = \log_2(s - r + 1)$, and if $t = 1$ then $K = K + 1$
6:         Create an expert $E_i$ by running an instance of the expert-algorithm to minimize $f_t(\cdot)$ during $I$, and set $w_{t-1,i} = 1/2$, $\widehat{w}_{t-1,i-1} = 1/2$, $L_{t-1,i} = 0$, $\widehat{L}_{t-1,i-1} = 0$
7:         Add $E_i$ to the set of active set: $\mathcal{A}_t = \mathcal{A}_t \cup \{E_i\}$
8:     **end for**
9:     Receive decision $\mathbf{x}_{t,i}$ from each expert $E_i \in \mathcal{A}_t$
10:     **for** $i = 0, \cdots, K - 1$ **do**
11:         **if** $i = 0$ **then**
12:             $\mathbf{z}_{t,0} = \mathbf{x}_{t,0}$
13:         **else**
14:             Calculate the weight $p_{t,i}$ and $\widehat{p}_{t,i-1}$ according to (2)
15:             Output the intermediate decision $\mathbf{z}_{t,i} = p_{t,i}\mathbf{x}_{t,i} + \widehat{p}_{t,i-1}\mathbf{z}_{t,i-1}$
16:             Observe the linearized loss $\ell_{t,i}, \widehat{\ell}_{t,i-1}$ in (3), and update $w_{t,i}, \widehat{w}_{t,i-1}$ in (4)
17:             Update $L_{t,i} = L_{t-1,i} + (\frac{1}{2} - \ell_{t,i})^2$ and $\widehat{L}_{t,i-1} = \widehat{L}_{t-1,i-1} + (\frac{1}{2} - \widehat{\ell}_{t,i-1})^2$
18:         **end if**
19:     **end for**
20:     Output the aggregated decision: $\mathbf{x}_t = \mathbf{z}_{t,K-1}$, and observe the loss function $f_t(\cdot)$
21:     Remove experts whose ending times are $t$ from $\mathcal{A}_t$
22: **end for**

---

expert's decision and the intermediate output, respectively:

$$\ell_{t,i} = \frac{\langle \nabla f_t(\mathbf{z}_{t,i}), \mathbf{x}_{t,i} - \mathbf{z}_{t,i} \rangle + GD}{2GD},$$
$$\widehat{\ell}_{t,i-1} = \frac{\langle \nabla f_t(\mathbf{z}_{t,i}), \mathbf{z}_{t,i-1} - \mathbf{z}_{t,i} \rangle + GD}{2GD}. \quad (3)$$

This normalization is performed because Adapt-ML-Prod requires the loss to be within $[0, 1]$. Moreover, we update $w_{t,i}$ and $\widehat{w}_{t,i-1}$ according to the rule:

$$w_{t,i} = \left( w_{t-1,i} \left( 1 + \eta_{t-1,i} \left( \frac{1}{2} - \ell_{t,i} \right) \right) \right)^{\frac{\eta_{t,i}}{\eta_{t-1,i}}},$$
$$\widehat{w}_{t,i-1} = \left( \widehat{w}_{t-1,i-1} \left( 1 + \widehat{\eta}_{t-1,i-1} \left( \frac{1}{2} - \widehat{\ell}_{t,i-1} \right) \right) \right)^{\frac{\widehat{\eta}_{t,i-1}}{\widehat{\eta}_{t-1,i-1}}}. \quad (4)$$

Finally, we use the last intermediate output as the final decision in Step 19.

In the following, we present the theoretical guarantee of IRESET.

**Theorem 2.** *Under Assumptions 1 and 2, for any segmentation $\mathcal{S}$, when functions are $\lambda$-strongly convex, by using OGD for strongly convex functions as the expert-algorithm, IRESET-OGD achieves*

$$\text{SW-REG}_T(\mathcal{S}) \leq \sum_{\mathcal{I} \in \mathcal{S}} 2\lceil \log_2 |\mathcal{I}| \rceil \left( 2\Xi + \frac{G^2}{\lambda} \ln |\mathcal{I}| \right)$$

*where $\Xi$ is defined as[1]*

$$\Xi = 2\Gamma GD \left( 2 + \frac{1}{\sqrt{\ln 2}} \right) + \frac{\Gamma^2 G^2}{2\lambda \ln 2},$$
$$\Gamma = 3 \ln 2 + \ln \left( 1 + \frac{1}{e}(1 + \ln(T+1)) \right). \quad (5)$$

*When functions are $\alpha$-exp-concave, by using ONS as the expert-algorithm, IRESET-ONS achieves*

$$\text{SW-REG}_T(\mathcal{S})$$
$$\leq \sum_{\mathcal{I} \in \mathcal{S}} 2\lceil \log_2 |\mathcal{I}| \rceil \left( 2\Xi' + 5d \left( \frac{1}{\alpha} + GD \right) \ln |\mathcal{I}| \right)$$

*where $\Xi' = 2\Gamma GD(2 + \frac{1}{\sqrt{\ln 2}}) + \frac{\Gamma^2}{2\alpha \ln 2}$.*

**Remark.** Theorem 2 demonstrates that by choosing appropriate expert-algorithms, IRESET achieves nearly-optimal switching regret $O(\frac{1}{\lambda} \sum_{\mathcal{I} \in \mathcal{S}} \log^2 |\mathcal{I}|)$ and $O(\frac{d}{\alpha} \sum_{\mathcal{I} \in \mathcal{S}} \log^2 |\mathcal{I}|)$ for $\lambda$-strongly convex and $\alpha$-exp-concave functions, respectively. Compared to the bounds obtained by strongly adaptive algorithms (Hazan & Seshadhri, 2009; Zhang et al., 2018b), our results provide an improvement from $O(\log T)$ to $O(\log |\mathcal{I}|)$ factor.

### 3.2. Our Refined Analysis

We present the analysis for strongly convex functions from Theorem 2 and defer the analysis for exp-concave functions to the appendix. The technical novelty of our analysis lies in two aspects: (i) for strongly convex functions, we specifically design a novel method to perform the recursion over the segment tree; and (ii) we propose a new proposition for the segment tree, which enables us to convert the tree-based switching regret into a formulation in terms of the segment lengths for these two types of functions.

We introduce the following assumption.

**Assumption 3.** *All the online functions $f_t(\cdot)$ are $\lambda$-strongly convex.*

---

[1] Following previous studies (Chernov & Vovk, 2010; Luo & Schapire, 2015), the double logarithmic factor in $\Xi$ can be treated as a constant, and thus does not affect the optimality.

To simplify our analysis, we define $\epsilon_t = \tilde{\epsilon}_k$ for all $t \in [\sigma_k, \sigma_{k+1} - 1]$, where

$$\tilde{\epsilon}_k = \operatorname*{argmin}_{\mathbf{x}^* \in \mathcal{X}} \sum_{t=\sigma_k}^{\sigma_{k+1}-1} f_t(\mathbf{x}^*).$$

First, we start with the meta-regret of IRESET w.r.t. the expert $E_{h(v)}$ and the intermediate output.

**Lemma 2.** *Under Assumptions 1, 2 and 3 for all vertices $v \in \mathcal{B}$ with $h(v) \neq 0$, we have*

$$\sum_{t=\blacktriangleleft(v)}^{\blacktriangleright(v)} f_t(\mathbf{z}_{t,h(v)})$$

$$\leq \min \left\{ \sum_{t=\blacktriangleleft(v)}^{\blacktriangleright(v)} f_t(\mathbf{x}_{t,h(v)}), \sum_{t=\blacktriangleleft(v)}^{\blacktriangleright(v)} f_t(\mathbf{z}_{t,h(v)-1}) \right\} + \Xi,$$

*where $\Xi$ is defined in (5).*

**Remark.** Our meta-algorithm can automatically exploit the property of strongly convex functions to achieve an $O(1)$ meta-regret, thus preserving the regret optimality. Furthermore, we do not require the meta-algorithm to support sleeping experts, thereby avoiding an additional $O(\log T)$ factor.

To bound the expert-regret, we choose OGD for strongly convex functions as the expert-algorithm and directly utilize its theoretical result (Shalev-Shwartz et al., 2011, Lemma 1).

**Lemma 3.** *Under Assumptions 1, 2 and 3, for all $v \in \mathcal{F}$, where $\mathcal{F}$ are fundamental vertices, we have*

$$\sum_{t=\blacktriangleleft(v)}^{\blacktriangleright(v)} f_t(\mathbf{x}_{t,h(v)}) \leq \sum_{t=\blacktriangleleft(v)}^{\blacktriangleright(v)} f_t(\epsilon_t) + \frac{G^2}{\lambda} \left( 1 + h(v) \ln 2 \right).$$

Combining Lemmas 2 and 3, we obtain the following recursive equations. For all $v \in \mathcal{F}$,

$$\sum_{t=\blacktriangleleft(v)}^{\blacktriangleright(v)} f_t(\mathbf{z}_{t,h(v)}) \leq \sum_{t=\blacktriangleleft(v)}^{\blacktriangleright(v)} f_t(\epsilon_t) + \Xi + \frac{G^2}{\lambda}(1+h(v)\ln 2). \tag{6}$$

And, for all $v \in \mathcal{R} \setminus \mathcal{F}$,

$$\sum_{t=\blacktriangleleft(v)}^{\blacktriangleright(v)} f_t(\mathbf{z}_{t,h(v)}) \leq \sum_{t=\blacktriangleleft(v)}^{\blacktriangleright(v)} f_t(\mathbf{z}_{t,h(v)-1}) + \Xi. \tag{7}$$

We utilize (6) and (7) to perform the recursion over the segment tree. This recursive procedure is employed to systematically bound the cumulative loss, $\sum_{t=\blacktriangleleft(v)}^{\blacktriangleright(v)} f_t(\mathbf{z}_{t,h(v)})$ for any relevant vertex $v \in \mathcal{R}$. (6) serves as the base case for fundamental vertices, directly bounding their performance against the optimal fixed decision $\epsilon_t$ with the sum

of the meta-regret and the expert-regret. (7) then provides the inductive step for non-fundamental relevant vertices, relating the loss at vertex $v$ to the aggregated loss from the level below (i.e., its children, corresponding to $h(v) - 1$) with the meta-regret accrued at depth $h(v)$. By applying these relationships iteratively up the segment tree from the fundamental descendants, we can derive a comprehensive bound on the learner's performance at any relevant node $v$, as formalized in Lemma 4.

**Lemma 4.** *Under Assumptions 1, 2 and 3, for all $v \in \mathcal{R}$, where $\mathcal{R}$ are relevant vertices, we have*

$$\sum_{t=\blacktriangleleft(v)}^{\blacktriangleright(v)} f_t(\mathbf{z}_{t,h(v)}) \leq \sum_{t=\blacktriangleleft(v)}^{\blacktriangleright(v)} f_t(\epsilon_t)$$

$$+ \sum_{q \in \mathcal{Q}(v)} \left( \Xi \sum_{k=0}^{h(v)-h(q)} 2^{-k} + \frac{G^2}{\lambda}(1 + h(q)\ln 2) \right)$$

*where $\mathcal{Q}(v)$ denotes the set of descendants of $v$ (including itself) that are contained in $\mathcal{F}$.*

By applying Lemma 4 with $v = r$, we derive a bound on the switching regret.

**Lemma 5.** *Under Assumptions 1, 2 and 3, we have*

$$\text{SW-REG}_T(\mathcal{S}) \leq \sum_{q \in \mathcal{F}} \left( 2\Xi + \frac{G^2}{\lambda}(1 + h(q)\ln 2) \right).$$

The above bound is based on the fundamental vertices within the segment tree. Therefore, we need to convert this into the switching regret in terms of the segment lengths. The original analysis of Pasteris et al. (2024, Lemma 4.7) cannot effectively handle strongly convex functions. Specifically, while their approach utilizes an inductive hypothesis to establish a connection between terms like $\sqrt{2^{h(v)}}$ and the segment length, this connection is not directly applicable to the $(h(v) \ln 2)$ term present in our Lemma 5. To address this issue, we propose the following useful proposition.

**Proposition 2.** *Let $\mathcal{F}_k$ be the set of fundamental vertices with respect to a given segmentation $\mathcal{S} = \{\mathcal{S}_1, \cdots, \mathcal{S}_k\}$ of the time horizon $[1, T]$. For each $k \in \{1, \cdots, \Phi\}$, let $\mathcal{S}_k = [\sigma_k, \sigma_{k+1} - 1]$ denote the $k$-th segment. Define the set $\mathcal{F}_k$ as:*

$$\mathcal{F}_k = \{v \in \mathcal{F} \mid \sigma_k \leq \blacktriangleleft(v) \text{ and } \blacktriangleright(v) < \sigma_{k+1}\}.$$

*For any $k \in \{1, \cdots, \Phi\}$, the number of fundamental vertices contained in the set $\mathcal{F}_k$ is at most*

$$|\mathcal{F}_k| \leq 2\lceil \log_2(\sigma_{k+1} - \sigma_k) \rceil.$$

*Proof.* As prior work (Pasteris et al., 2024, Lemma 4.7) pointed out, for any depth $i \in \{0, 1, \cdots, h(r)\}$ in the segment tree, there are at most two distinct vertices $v, v' \in \mathcal{F}_k$

with $h(v) = h(v') = i$. Therefore, we can partition $\mathcal{F}_k$ into two disjoint sets $\mathcal{U}_k$ and $\mathcal{V}_k$, such that for all $i \in \{0, 1, \cdots, h(r)\}$ there exists at most one element $v$ of $\mathcal{U}_k$ with $h(v) = i$ and at most one element $v'$ of $\mathcal{V}_k$ with $h(v') = i$.

Since the largest interval maintained by any vertex in either set does not exceed $\sigma_{k+1} - \sigma_k$, the number of fundamental vertices contained is at most $\lceil \log_2(\sigma_{k+1} - \sigma_k) \rceil$ for both sets $\mathcal{U}_k$ and $\mathcal{V}_k$. $\qquad\square$

**Remark.** Proposition 2 reveals that each segment $\mathcal{S}_k$ contains at most $2\lceil \log_2(\sigma_{k+1} - \sigma_k) \rceil$ fundamental vertices, which allows for a direct conversion of the tree-based switching regret into a formulation in terms of segment lengths. Combining Proposition 2 with Lemma 5, we have

$$
\begin{aligned}
&\text{SW-Reg}(\mathcal{S}) \\
&\leq \sum_{k=1}^{\Phi} \sum_{q \in \mathcal{F}_k} \left( 2\Xi + \frac{G^2}{\lambda}(1 + h(q)\ln 2) \right) \\
&\leq \sum_{k=1}^{\Phi} 2\lceil \log_2(\sigma_{k+1} - \sigma_k) \rceil \left( 2\Xi + \frac{G^2}{\lambda}\ln(\sigma_{k+1} - \sigma_k) \right) \\
&= O\left( \frac{1}{\lambda} \sum_{\mathcal{I} \in \mathcal{S}} \log^2 |\mathcal{I}| \right)
\end{aligned}
$$

where the second inequality is due to the fact that $2^{h(q)} \leq \sigma_{k+1} - \sigma_k$ for all $q \in \mathcal{F}_k$.

### 3.3. A Universal Strategy

Combined with the result of Pasteris et al. (2024), switching regret bounds have been successfully established for general convex, exp-concave, and strongly convex functions, respectively. However, the associated algorithms lack universality and can only handle one type of convex functions, which drives us to develop a universal strategy for switching regret.

Before stating our algorithm, we briefly review the related work on universal online learning. For static regret, pioneering work is adaptive online gradient descent (AOGD) (Bartlett et al., 2008), which interpolates between general convex functions and strongly convex functions. Another milestone for universal OCO is MetaGrad (van Erven & Koolen, 2016; van Erven et al., 2021), which maintains multiple experts for different types of functions and combines them via a meta-algorithm. Wang et al. (2019) further prove the optimal regret for strongly convex functions and extended it to support smoothness (Wang et al., 2020). Zhang et al. (2022) introduce a simple yet universal strategy that allows experts to operate the original function. However, for convex functions, this method does not achieve optimal gradient variation bounds. To address this gap, Yan et al. (2023) propose a multi-layer framework, which is later

refined by Yan et al. (2024). For adaptive regret, Zhang et al. (2021) propose a universal algorithm for minimizing the adaptive regret of convex functions.

In this subsection, we go one step further by developing a universal algorithm for achieving switching regret for three types of convex functions simultaneously. By employing an existing universal algorithm for static regret, i.e., Maler (Wang et al., 2019), as the expert-algorithm for IRESET, our algorithm can adapt to the function type and its associated unknown parameters. Therefore, we can create multiple experts by running instances of Maler, and employ IRESET to aggregate these experts' decisions.

Finally, we provide the theoretical guarantee for IRESET with Maler expert.

**Theorem 3.** *Under Assumptions 1 and 2, for any segmentation $\mathcal{S}$, by employing Maler as the expert-algorithm, IRESET-Maler is able to achieve $O(\frac{d}{\alpha} \sum_{\mathcal{I} \in \mathcal{S}} \log^2 |\mathcal{I}|)$, $O(\frac{1}{\lambda} \sum_{\mathcal{I} \in \mathcal{S}} \log^2 |\mathcal{I}|)$ and $O(\sum_{\mathcal{I} \in \mathcal{S}} \sqrt{|\mathcal{I}|})$ for $\alpha$-exp-concave, $\lambda$-strongly convex, and general convex functions.*

**Remark.** Theorem 3 demonstrates that IRESET-Maler is able to deliver switching regret bounds for three types of convex functions simultaneously. It matches the optimal switching regret bounds for general convex functions (Pasteris et al., 2024), while also attaining the nearly-optimal logarithmic switching regret for exp-concave or strongly convex functions. Crucially, this is achieved without prior knowledge of the modulus of strong convexity or exp-concavity.

**Clarification.** We would like to emphasize that although our results still exhibit an $O(\log |\mathcal{I}|)$ gap from the lower bound established in Theorem 1, our work makes an important contribution to the analysis of switching regret. In future work, we will investigate whether this additional factor can be eliminated, thereby achieving optimal switching regret for strongly convex functions and exp-concave functions.

**Discussions on Dynamic Regret.** Many studies (Zhang et al., 2018a; Wang et al., 2024; Zhao et al., 2024; Wan, 2025; Yang et al., 2026) have been devoted to addressing changing environments, where dynamic regret is commonly used. The prior work of Pasteris et al. (2024) has indeed provided the optimal dynamic regret for general convex functions. However, their analysis crucially relies on using OGD as the expert-algorithm and exploiting $O(\sqrt{T}(1 + P_T))$. We remark that extending this to strongly convex or exp-concave functions presents fundamental difficulties. The key challenge is that OGD/ONS algorithms for strongly convex or exp-concave functions do not admit dynamic regret bounds that depend on the path length of the comparator sequence. This prevents the per-segment expert-regret from being bounded in terms of the path length, making it difficult to accommodate changing comparators.

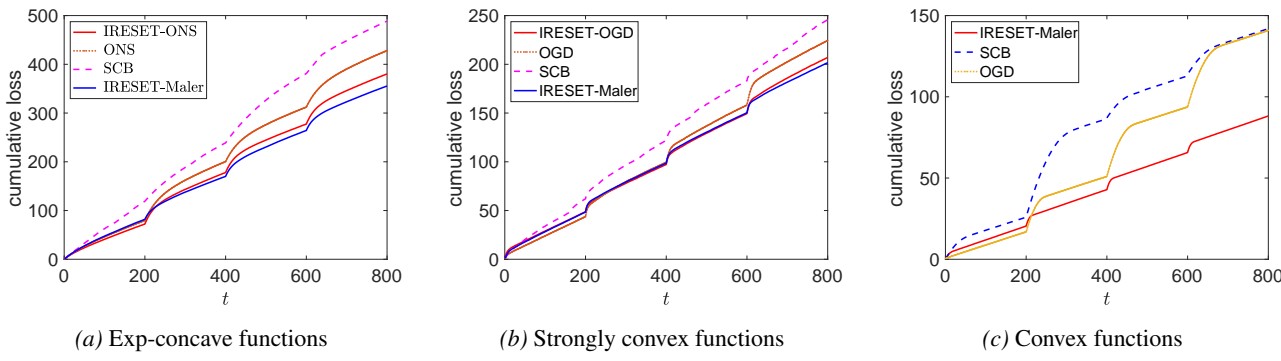

*(a)* Exp-concave functions      *(b)* Strongly convex functions      *(c)* Convex functions

*Figure 3.* Cumulative losses of different methods versus the number of iterations.

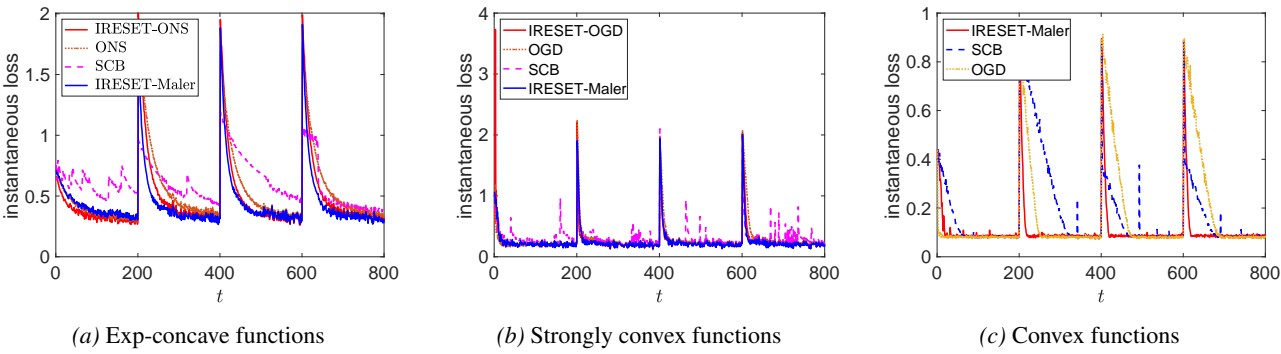

*(a)* Exp-concave functions      *(b)* Strongly convex functions      *(c)* Convex functions

*Figure 4.* Instantaneous losses of different methods versus the number of iterations.

## 4. Experiments

In this section, we conduct empirical experiments to validate the effectiveness of our proposed methods.

We conduct experiments on the ijcnn1 dataset from LIB-SVM Data (Chang & Lin, 2011; Prokhorov, 2001), where the dimension of features is $d = 22$. We consider the following online classification problem. In each round $t \in [T]$, the online learner chooses a decision $\mathbf{x}_t \in \mathcal{X}$. After submitting the decision, the online learner receives a batch of data samples $\{(x_t^{(i)}, y_t^{(i)})\}_{i=1}^m$ which are sampled from the dataset, where $x_t^{(i)}$ is the feature vector of the $i$-th sample, and $y_t^{(i)}$ is the corresponding label. The learner can evaluate the model by the online convex loss $f_t(\mathbf{x}_t)$ and update the decision for the next round. To simulate the changing environment, we flip the labels every 200 rounds. For the function setup, we consider three types of online convex functions, including a logistic loss (exp-concave), a regularized hinge loss (strongly-convex), and a absolute loss (general convex). We compare the performance of our proposed methods with existing algorithms, including OGD, SCB (Jun et al., 2017), and RESET (Pasteris et al., 2024).

We repeat the experiments for five times and record the results in Figure 3 and 4. As shown in Figure 3, the proposed IRESET-based methods consistently achieve smaller cumulative losses than the baselines under exp-concave, strongly convex, and general convex losses. Figure 4 further shows that after each label flip, which occurs every 200 rounds, our methods recover quickly and maintain relatively low instantaneous losses. These results demonstrate the effectiveness of IRESET in changing environments, while IRESET-Maler also exhibits competitive performance, i.e., universality, across different function classes.

## 5. Conclusion

In this paper, we investigate OCO with the switching regret. For exp-concave or strongly convex functions, we propose a novel meta-algorithm, termed IRESET, which sequentially aggregates experts' decisions over DGC intervals using a meta-algorithm with second-order bounds. Our analysis demonstrates that IRESET with appropriate expert-algorithms achieve nearly-optimal logarithmic switching regret for these two types of convex functions. This is enabled by a novel analysis involving recursive equations over the segment trees and a structural property of fundamental vertices. Furthermore, we develop IRESET-Maler, a universal algorithm that attains switching regret for general convex, exp-concave, and strongly convex functions simultaneously, without knowledge of the function types.

## Acknowledgements

This work was partially supported by NSFC (U23A20382, 62361146852), the "111 Center" (No. B26023), and the Collaborative Innovation Center of Novel Software Technology and Industrialization.

## Impact Statement

This paper presents work whose goal is to advance the field of Machine Learning. There are many potential societal consequences of our work, none which we feel must be specifically highlighted here.

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

# A. Proofs

In this section, we present the proofs of all theorems and lemmas.

## A.1. Proof of Theorem 1

Assuming the segmentation $\mathcal{S}$ is known a priori, we can employ the OGD or ONS algorithm, restarting the chosen algorithm at the beginning of each segment. This strategy effectively decomposes the problem, allowing each segment to be treated as an independent static regret minimization task.

Consequently, by applying the established static regret lower bounds for three types of convex functions (Ordentlich & Cover, 1998; Abernethy et al., 2008) to each segment individually, we can directly derive a lower bound for the switching regret by aggregating these per-segment bounds.

## A.2. Proof of Lemma 2

According to the theoretical guarantee of Adapt-ML-Prod (Gaillard et al., 2014, Corollary 4), for all vertices $v \in \mathcal{B}$ with $h(v) \neq 0$, we have

$$\sum_{t=\blacktriangleleft(v)}^{\blacktriangleright(v)} \frac{1}{2} - \sum_{t=\blacktriangleleft(v)}^{\blacktriangleright(v)} \ell_{t,h(v)} \leq \frac{\Gamma}{\sqrt{\ln 2}} \sqrt{1 + \sum_{t=\blacktriangleleft(v)}^{\blacktriangleright(v)} \left( \frac{1}{2} - \ell_{t,h(v)} \right)^2} + 2\Gamma \tag{8}$$

$$\sum_{t=\blacktriangleleft(v)}^{\blacktriangleright(v)} \frac{1}{2} - \sum_{t=\blacktriangleleft(v)}^{\blacktriangleright(v)} \widehat{\ell}_{t,h(v)-1} \leq \frac{\Gamma}{\sqrt{\ln 2}} \sqrt{1 + \sum_{t=\blacktriangleleft(v)}^{\blacktriangleright(v)} \left( \frac{1}{2} - \widehat{\ell}_{t,h(v)-1} \right)^2} + 2\Gamma \tag{9}$$

where $\Gamma$ is defined as

$$\Gamma = 3\ln 2 + \ln\left(1 + \frac{1}{e}(1 + \ln(T+1))\right) = O(\log\log T).$$

According to the definitions of the linearized loss in (3), the definitions of $\ell_{t,h(v)}$ and $\widehat{\ell}_{t,h(v)-1}$ are

$$\ell_{t,h(v)} = \frac{\langle \nabla f_t(\mathbf{z}_{t,h(v)}), \mathbf{x}_{t,h(v)} - \mathbf{z}_{t,h(v)} \rangle + GD}{2GD},$$

$$\widehat{\ell}_{t,h(v)-1} = \frac{\langle \nabla f_t(\mathbf{z}_{t,h(v)}), \mathbf{z}_{t,h(v)-1} - \mathbf{z}_{t,h(v)} \rangle + GD}{2GD}.$$

Therefore, the meta-loss in (8) and (9) is

$$\ell_t = p_{t,i}\ell_{t,i} + \widehat{p}_{t,i-1}\widehat{\ell}_{t,i-1} = \frac{1}{2}.$$

Combining the definition of $\ell_{t,h(v)}$ with (8), we arrive at

$$\sum_{t=\blacktriangleleft(v)}^{\blacktriangleright(v)} \langle \nabla f_t(\mathbf{z}_{t,h(v)}), \mathbf{z}_{t,h(v)} - \mathbf{x}_{t,h(v)} \rangle$$

$$\leq \frac{\Gamma}{\sqrt{\ln 2}} \sqrt{4G^2D^2 + \sum_{t=\blacktriangleleft(v)}^{\blacktriangleright(v)} \langle \nabla f_t(\mathbf{z}_{t,h(v)}), \mathbf{z}_{t,h(v)} - \mathbf{x}_{t,h(v)} \rangle^2} + 4\Gamma GD$$

$$\leq 2\Gamma GD \left(2 + \frac{1}{\sqrt{\ln 2}}\right) + \frac{\Gamma^2 G^2}{2\lambda \ln 2} + \frac{\lambda}{2G^2} \sum_{t=\blacktriangleleft(v)}^{\blacktriangleright(v)} \langle \nabla f_t(\mathbf{z}_{t,h(v)}), \mathbf{z}_{t,h(v)} - \mathbf{x}_{t,h(v)} \rangle^2 \tag{10}$$

$$\leq 2\Gamma GD \left(2 + \frac{1}{\sqrt{\ln 2}}\right) + \frac{\Gamma^2 G^2}{2\lambda \ln 2} + \frac{\lambda}{2} \sum_{t=\blacktriangleleft(v)}^{\blacktriangleright(v)} \|\mathbf{z}_{t,h(v)} - \mathbf{x}_{t,h(v)}\|^2$$

where the second inequality is due to $\sqrt{a+b} \le \sqrt{a} + \sqrt{b}$, $\sqrt{ab} \le \frac{a}{2} + \frac{b}{2}$, and the last step is due to the bounded gradients in Assumption 1. Based on the strong convexity in Definition 1, we have

$$\sum_{t=\blacktriangleleft(v)}^{\blacktriangleright(v)} f_t(\mathbf{z}_{t,h(v)}) - \sum_{t=\blacktriangleleft(v)}^{\blacktriangleright(v)} f_t(\mathbf{x}_{t,h(v)}) \le \underbrace{2\Gamma GD\left(2 + \frac{1}{\sqrt{\ln 2}}\right) + \frac{\Gamma^2 G^2}{2\lambda \ln 2}}_{:=\Xi}.$$

Similarly, combining the definition of $\hat{\ell}_{t,h(v)-1}$ with (9), we have

$$\sum_{t=\blacktriangleleft(v)}^{\blacktriangleright(v)} f_t(\mathbf{z}_{t,h(v)}) - \sum_{t=\blacktriangleleft(v)}^{\blacktriangleright(v)} f_t(\mathbf{z}_{t,h(v)-1}) \le \Xi.$$

Combining the above inequalities, we finish the proof.

### A.3. Proof of Lemma 4

We use inductive hypothesis to prove it. First, we define

$$\phi_k := \sum_{i=0}^{k} 2^{-i}.$$

Thus, the inductive hypothesis is

$$\sum_{t=\blacktriangleleft(v)}^{\blacktriangleright(v)} f_t(\mathbf{z}_{t,h(v)}) \le \sum_{t=\blacktriangleleft(v)}^{\blacktriangleright(v)} f_t(\epsilon_t) + \sum_{q \in \mathcal{Q}(v)} \left(\phi_{h(v)-h(q)} \cdot \Xi + \frac{G^2}{\lambda}(1 + h(q)\ln 2)\right) \tag{11}$$

for all $v \in \mathcal{R}$. If $v \in \mathcal{F}$, we have $\mathcal{Q}(v) = \{v\}$, and (11) holds. If $v \in \mathcal{R} \setminus \mathcal{F}$, let $\triangleleft(v), \triangleright(v) \in \mathcal{R}$ be the left and right child of vertex $v$. According to the property of the segment tree, we have

$$\sum_{t=\blacktriangleleft(v)}^{\blacktriangleright(v)} f_t(\mathbf{z}_{t,h(v)-1}) = \sum_{t=\blacktriangleleft(\triangleleft(v))}^{\blacktriangleright(\triangleleft(v))} f_t(\mathbf{z}_{t,h(\triangleleft(v))}) + \sum_{t=\blacktriangleleft(\triangleright(v))}^{\blacktriangleright(\triangleright(v))} f_t(\mathbf{z}_{t,h(\triangleright(v))}). \tag{12}$$

Lemma 2 implies

$$\begin{aligned}
\sum_{t=\blacktriangleleft(v)}^{\blacktriangleright(v)} f_t(\mathbf{z}_{t,h(v)}) &\le \sum_{t=\blacktriangleleft(v)}^{\blacktriangleright(v)} f_t(\mathbf{z}_{t,h(v)-1}) + \Xi \\
&= \sum_{t=\blacktriangleleft(\triangleleft(v))}^{\blacktriangleright(\triangleleft(v))} f_t(\mathbf{z}_{t,h(\triangleleft(v))}) + \sum_{t=\blacktriangleleft(\triangleright(v))}^{\blacktriangleright(\triangleright(v))} f_t(\mathbf{z}_{t,h(\triangleright(v))}) + \Xi \\
&= \sum_{t=\blacktriangleleft(\triangleleft(v))}^{\blacktriangleright(\triangleleft(v))} f_t(\mathbf{z}_{t,h(\triangleleft(v))}) + \sum_{t=\blacktriangleleft(\triangleright(v))}^{\blacktriangleright(\triangleright(v))} f_t(\mathbf{z}_{t,h(\triangleright(v))}) + \sum_{q \in \mathcal{Q}(v)} 2^{h(q)-h(v)} \cdot \Xi
\end{aligned}$$

where the last step is due to

$$\sum_{q \in \mathcal{Q}(v)} 2^{h(q)} = 2^{h(v)}.$$

Applying the inductive hypothesis to the vertex $\triangleleft(v)$ and $\triangleright(v)$, we have

$$\begin{aligned}
\sum_{t=\blacktriangleleft(v)}^{\blacktriangleright(v)} f_t(\mathbf{z}_{t,h(v)}) &\le \sum_{t=\blacktriangleleft(v)}^{\blacktriangleright(v)} f_t(\epsilon_t) + \sum_{q \in \mathcal{Q}(\triangleleft(v))} \left(\phi_{h(\triangleleft(v))-h(q)} \cdot \Xi + \frac{G^2}{\lambda}(1 + h(q)\ln 2)\right) \\
&\quad + \sum_{q \in \mathcal{Q}(\triangleright(v))} \left(\phi_{h(\triangleright(v))-h(q)} \cdot \Xi + \frac{G^2}{\lambda}(1 + h(q)\ln 2)\right) + \sum_{q \in \mathcal{Q}(v)} 2^{h(q)-h(v)} \cdot \Xi.
\end{aligned}$$

We know that $\mathcal{Q}(v) = \mathcal{Q}(\triangleright(v)) \cup \mathcal{Q}(\triangleleft(v))$, and for all $q \in \mathcal{Q}(v)$ we have

$$\phi_{h(v)-h(q)} = \phi_{h(\triangleright(v))-h(q)} + 2^{h(q)-h(v)} = \phi_{h(\triangleleft(v))-h(q)} + 2^{h(q)-h(v)}.$$

Combining them together, we show that the inductive hypothesis holds for $v \in \mathcal{R} \setminus \mathcal{F}$. Therefore, we have proved that the inductive hypothesis always holds for all $v \in \mathcal{R}$, and finish the proof.

### A.4. Proof of Lemma 5

Based on Lemma 4, we notice that

$$\sum_{k=0}^{h(v)-h(q)} 2^{-i} \leq \sum_{k=0}^{+\infty} 2^{-k} = 2.$$

Applying Lemma 4 by setting $v = r$, we attain

$$\sum_{t=\blacktriangleleft(r)}^{\blacktriangleright(r)} f_t(\mathbf{z}_{t,h(r)}) \leq \sum_{t=\blacktriangleleft(r)}^{\blacktriangleright(r)} f_t(\epsilon_t) + \sum_{q \in \mathcal{Q}(r)} \left( 2\Xi + \frac{G^2}{\lambda}(1 + h(q)\ln 2) \right)$$

$$\leq \sum_{t=\blacktriangleleft(r)}^{\blacktriangleright(r)} f_t(\epsilon_t) + 2\left( \Xi + \frac{G^2}{\lambda} \right) \sum_{q \in \mathcal{Q}(r)} \ln 2^{h(q)+2}$$

Since $\blacktriangleleft(r) = 1$, $\blacktriangleright(r) = T$, and $\mathbf{z}_{t,h(r)} = \mathbf{x}_t$, we have

$$\text{SW-Reg}(\mathcal{S}) \leq \sum_{q \in \mathcal{F}} \left( 2\Xi + \frac{G^2}{\lambda}(1 + h(q)\ln 2) \right).$$

We finish the proof.

### A.5. Proof of Theorem 2

In Section 3.2, we provide a refined analysis, which proves a nearly-optimal switching regret for $\lambda$-strongly convex functions. In this section, we proceed to handle $\alpha$-exp-concave functions, where the analysis is similar. For convenience, we introduce the following assumption.

**Assumption 4.** *All the online functions $f_t(\cdot)$ are $\alpha$-exp-concave.*

First, we start with the meta-regret.

**Lemma 6.** *Under Assumptions 1, 2 and 4 for all vertices $v \in \mathcal{B}$ with $h(v) \neq 0$, we have*

$$\sum_{t=\blacktriangleleft(v)}^{\blacktriangleright(v)} f_t(\mathbf{z}_{t,h(v)}) \leq \min \left\{ \sum_{t=\blacktriangleleft(v)}^{\blacktriangleright(v)} f_t(\mathbf{x}_{t,h(v)}), \sum_{t=\blacktriangleleft(v)}^{\blacktriangleright(v)} f_t(\mathbf{z}_{t,h(v)-1}) \right\} + \Xi',$$

*where $\Xi' = 2\Gamma G D(2 + \frac{1}{\sqrt{\ln 2}}) + \frac{\Gamma^2}{2\alpha \ln 2}$, and $\Gamma$ is defined in (5).*

To bound the expert-regret for $\alpha$-exp-concave functions, we choose ONS as the expert-algorithm and directly utilize its theoretical result (Hazan et al., 2007, Theorem 2).

**Lemma 7.** *Under Assumptions 1, 2 and 4, for all $v \in \mathcal{F}$, we have*

$$\sum_{t=\blacktriangleleft(v)}^{\blacktriangleright(v)} f_t(\mathbf{x}_{t,h(v)}) \leq \sum_{t=\blacktriangleleft(v)}^{\blacktriangleright(v)} f_t(\epsilon_t) + 5d\left( \frac{1}{\alpha} + GD \right) h(v)\ln 2.$$

Combining Lemmas 6 and 7, we obtain the following recursive equations:

$$\sum_{t=\blacktriangleleft(v)}^{\blacktriangleright(v)} f_t(\mathbf{z}_{t,h(v)}) \leq \sum_{t=\blacktriangleleft(v)}^{\blacktriangleright(v)} f_t(\boldsymbol{\epsilon}_t) + \Xi' + 5d \left( \frac{1}{\alpha} + GD \right) h(v) \ln 2, \text{ for all } v \in \mathcal{F}, \tag{13}$$

$$\sum_{t=\blacktriangleleft(v)}^{\blacktriangleright(v)} f_t(\mathbf{z}_{t,h(v)}) \leq \sum_{t=\blacktriangleleft(v)}^{\blacktriangleright(v)} f_t(\mathbf{z}_{t,h(v)-1}) + \Xi', \text{ for all } v \in \mathcal{R} \setminus \mathcal{F}. \tag{14}$$

We utilize (13) and (14) to perform the recursion over the segment tree.

**Lemma 8.** *Under Assumptions 1, 2 and 4, for all $v \in \mathcal{R}$, we have*

$$\sum_{t=\blacktriangleleft(v)}^{\blacktriangleright(v)} f_t(\mathbf{z}_{t,h(v)}) \leq \sum_{t=\blacktriangleleft(v)}^{\blacktriangleright(v)} f_t(\boldsymbol{\epsilon}_t) + \sum_{q \in \mathcal{Q}(v)} \left( \Xi' \sum_{k=0}^{h(v)-h(q)} 2^{-k} + 5d \left( \frac{1}{\alpha} + GD \right) h(v) \ln 2 \right)$$

*where $\mathcal{Q}(v)$ denotes the set of descendants of $v$ (including itself) that are contained in $\mathcal{F}$.*

We omit the proof because it is the same as that of Lemma 4. By applying Lemma 8 with $v = r$, we derive a bound on the switching regret.

**Lemma 9.** *Under Assumptions 1, 2 and 4, we have*

$$\text{SW-REG}(\mathcal{S}) \leq \sum_{q \in \mathcal{F}} \left( 2\Xi' + 5d \left( \frac{1}{\alpha} + GD \right) h(q) \ln 2 \right).$$

Combining Lemma 9 with Proposition 2, we have

$$\begin{aligned}
\text{SW-REG}(\mathcal{S}) &\leq \sum_{k=1}^{\Phi} \sum_{q \in \mathcal{F}_k} \left( 2\Xi' + 5d \left( \frac{1}{\alpha} + GD \right) h(q) \ln 2 \right) \\
&\leq \sum_{k=1}^{\Phi} 2 \lceil \log_2(\sigma_{k+1} - \sigma_k) \rceil \left( 2\Xi' + 5d \left( \frac{1}{\alpha} + GD \right) \ln(\sigma_{k+1} - \sigma_k) \right).
\end{aligned} \tag{15}$$

Finally, we complete the proof by using $\mathcal{S} = \{\sigma_1, \cdots, \sigma_{\Phi+1}\}$.

### A.6. Proof of Lemma 6

The analysis is similar to that of Lemma 2. (10) can be rewritten as

$$\begin{aligned}
&\sum_{t=\blacktriangleleft(v)}^{\blacktriangleright(v)} \langle \nabla f_t(\mathbf{z}_{t,h(v)}), \mathbf{z}_{t,h(v)} - \mathbf{x}_{t,h(v)} \rangle \\
&\leq \frac{\Gamma}{\sqrt{\ln 2}} \sqrt{4G^2 D^2 + \sum_{t=\blacktriangleleft(v)}^{\blacktriangleright(v)} \langle \nabla f_t(\mathbf{z}_{t,h(v)}), \mathbf{z}_{t,h(v)} - \mathbf{x}_{t,h(v)} \rangle^2 + 4\Gamma GD} \\
&\leq 2\Gamma GD \left( 2 + \frac{1}{\sqrt{\ln 2}} \right) + \frac{\Gamma^2}{2\alpha \ln 2} + \frac{\alpha}{2} \sum_{t=\blacktriangleleft(v)}^{\blacktriangleright(v)} \langle \nabla f_t(\mathbf{z}_{t,h(v)}), \mathbf{z}_{t,h(v)} - \mathbf{x}_{t,h(v)} \rangle^2
\end{aligned} \tag{16}$$

where the last step is due to $\sqrt{a+b} \leq \sqrt{a} + \sqrt{b}$ and $\sqrt{ab} \leq \frac{a}{2} + \frac{b}{2}$.

According to Lemma 1, (16) implies

$$\sum_{t=\blacktriangleleft(v)}^{\blacktriangleright(v)} f_t(\mathbf{z}_{t,h(v)}) - \sum_{t=\blacktriangleleft(v)}^{\blacktriangleright(v)} f_t(\mathbf{x}_{t,h(v)}) \leq \underbrace{2\Gamma GD \left( 2 + \frac{1}{\sqrt{\ln 2}} \right) + \frac{\Gamma^2}{2\alpha \ln 2}}_{:=\Xi'}.$$

Similarly, we have

$$\sum_{t=\blacktriangleleft(v)}^{\blacktriangleright(v)} f_t(\mathbf{z}_{t,h(v)}) - \sum_{t=\blacktriangleleft(v)}^{\blacktriangleright(v)} f_t(\mathbf{z}_{t,h(v)-1}) \le \Xi'.$$

Combining the above inequalities, we finish the proof.

## A.7. Proof of Theorem 3

We provide proofs for general convex functions, $\lambda$-strongly convex functions, and $\alpha$-exp-concave functions, respectively.

### A.7.1. ANALYSIS FOR GENERAL CONVEX FUNCTIONS

First, we start with the meta-regret. (10) can be rewritten as

$$
\begin{aligned}
\sum_{t=\blacktriangleleft(v)}^{\blacktriangleright(v)} f_t(\mathbf{z}_{t,h(v)}) - \sum_{t=\blacktriangleleft(v)}^{\blacktriangleright(v)} f_t(\mathbf{x}_{t,h(v)}) &\le \sum_{t=\blacktriangleleft(v)}^{\blacktriangleright(v)} \langle \nabla f_t(\mathbf{z}_{t,h(v)}), \mathbf{z}_{t,h(v)} - \mathbf{x}_{t,h(v)} \rangle \\
&\le \frac{\Gamma}{\sqrt{\ln 2}} \sqrt{4G^2 D^2 + \sum_{t=\blacktriangleleft(v)}^{\blacktriangleright(v)} \langle \nabla f_t(\mathbf{z}_{t,h(v)}), \mathbf{z}_{t,h(v)} - \mathbf{x}_{t,h(v)} \rangle^2 + 4\Gamma GD} \\
&\le 2\Gamma GD \left( 2 + \frac{1}{\sqrt{\ln 2}} \right) + \frac{\Gamma GD}{\sqrt{\ln 2}} \sqrt{2^{h(v)}}
\end{aligned}
\tag{17}
$$

where the last step is $\sqrt{a+b} \le \sqrt{a} + \sqrt{b}$, $\sqrt{ab} \le \frac{a}{2} + \frac{b}{2}$ and Assumptions 1 and 2. Similarly, we also have

$$\sum_{t=\blacktriangleleft(v)}^{\blacktriangleright(v)} f_t(\mathbf{z}_{t,h(v)}) - \sum_{t=\blacktriangleleft(v)}^{\blacktriangleright(v)} f_t(\mathbf{z}_{t,h(v)-1}) \le 2\Gamma GD \left( 2 + \frac{1}{\sqrt{\ln 2}} \right) + \frac{\Gamma GD}{\sqrt{\ln 2}} \sqrt{2^{h(v)}}.$$

Combing the above inequality with (17), we have

$$
\begin{aligned}
\sum_{t=\blacktriangleleft(v)}^{\blacktriangleright(v)} f_t(\mathbf{z}_{t,h(v)}) \le \min \left\{ \sum_{t=\blacktriangleleft(v)}^{\blacktriangleright(v)} f_t(\mathbf{x}_{t,h(v)}), \sum_{t=\blacktriangleleft(v)}^{\blacktriangleright(v)} f_t(\mathbf{z}_{t,h(v)-1}) \right\} \\
+ \Gamma GD \left( 4 + \frac{3}{\sqrt{\ln 2}} \right) \sqrt{2^{h(v)}}
\end{aligned}
\tag{18}
$$

To bound the expert-regret, we can directly use the theoretical guarantee of Maler (Wang et al., 2019, Theorem 1).

$$\sum_{t=\blacktriangleleft(v)}^{\blacktriangleright(v)} f_t(\mathbf{x}_{t,h(v)}) \le \sum_{t=\blacktriangleleft(v)}^{\blacktriangleright(v)} f_t(\boldsymbol{\epsilon}_t) + \left( 2\ln 3 + \frac{3}{2} \right) GD\sqrt{2^{h(v)}}.
\tag{19}$$

Combining (18) and (19), we derive the following recursive equations:

$$\sum_{t=\blacktriangleleft(v)}^{\blacktriangleright(v)} f_t(\mathbf{z}_{t,h(v)}) \le \sum_{t=\blacktriangleleft(v)}^{\blacktriangleright(v)} f_t(\boldsymbol{\epsilon}_t) + \left( \gamma + \Gamma GD \left( 4 + \frac{3}{\sqrt{\ln 2}} \right) \right) \sqrt{2^{h(v)}}, \text{ for all } v \in \mathcal{F},
\tag{20}$$

$$\sum_{t=\blacktriangleleft(v)}^{\blacktriangleright(v)} f_t(\mathbf{z}_{t,h(v)}) \le \sum_{t=\blacktriangleleft(v)}^{\blacktriangleright(v)} f_t(\mathbf{z}_{t,h(v)-1}) + \Gamma GD \left( 4 + \frac{3}{\sqrt{\ln 2}} \right) \sqrt{2^{h(v)}}, \text{ for all } v \in \mathcal{R} \setminus \mathcal{F}.
\tag{21}$$

where

$$\gamma = \left( 2\ln 3 + \frac{3}{2} \right) GD.
\tag{22}$$

We utilize (20) and (21) to perform the recursion. We can directly utilize Pasteris et al. (2024, Lemma 4.5) to obtain, for all $v \in \mathcal{R}$, we have

$$\sum_{t=\blacktriangleleft(v)}^{\blacktriangleright(v)} f_t(\mathbf{z}_{t,h(v)}) \le \sum_{t=\blacktriangleleft(v)}^{\blacktriangleright(v)} f_t(\boldsymbol{\epsilon}_t) + \sum_{q \in \mathcal{Q}(v)} \left( \gamma + \Gamma G D \left( 4 + \frac{3}{\sqrt{\ln 2}} \right) \sum_{k=0}^{h(v)-h(q)} \sqrt{2^{-k}} \right) \sqrt{2^{h(q)}}$$

where $\mathcal{Q}(v)$ denotes the set of descendants of $v$ (including itself) that are contained in $\mathcal{F}$. Next, by applying the above inequality with $v = r$, we have

$$\text{SW-REG}(\mathcal{S}) \le \left( \gamma + 4\Gamma G D \left( 4 + \frac{3}{\sqrt{\ln 2}} \right) \right) \sum_{q \in \mathcal{F}} \sqrt{2^{h(q)}}.$$

Finally, we can use Pasteris et al. (2024, Lemma 4.7) to finish the proof.

### A.7.2. ANALYSIS FOR STRONGLY CONVEX FUNCTIONS AND EXP-CONCAVE FUNCTIONS

The analysis is similar to that of Theorem 2. To bound the meta-regret, we can use Lemma 2 to obtain

$$\sum_{t=\blacktriangleleft(v)}^{\blacktriangleright(v)} f_t(\mathbf{z}_{t,h(v)}) \le \min \left\{ \sum_{t=\blacktriangleleft(v)}^{\blacktriangleright(v)} f_t(\mathbf{x}_{t,h(v)}), \sum_{t=\blacktriangleleft(v)}^{\blacktriangleright(v)} f_t(\mathbf{z}_{t,h(v)-1}) \right\} + \Xi,$$

where $\Xi$ is defined in (5). According to the theoretical guarantee of Maler (Wang et al., 2019, Theorem 1), we have

$$\sum_{t=\blacktriangleleft(v)}^{\blacktriangleright(v)} f_t(\mathbf{x}_{t,h(v)}) \le \sum_{t=\blacktriangleleft(v)}^{\blacktriangleright(v)} f_t(\boldsymbol{\epsilon}_t) + \left( 10GD + \frac{9G^2}{2\lambda} \right) (\gamma' + 1 + h(v) \ln 2) \tag{23}$$

where

$$\gamma' = 2 \ln \left( \frac{\sqrt{3}}{2} \log_2(s - r + 1) + 3\sqrt{3} \right) = O(\log \log T). \tag{24}$$

Combining Lemma 2 and (23), we have

$$\sum_{t=\blacktriangleleft(v)}^{\blacktriangleright(v)} f_t(\mathbf{z}_{t,h(v)}) \le \sum_{t=\blacktriangleleft(v)}^{\blacktriangleright(v)} f_t(\boldsymbol{\epsilon}_t) + \Xi + \left( 10GD + \frac{9G^2}{2\lambda} \right) (\gamma' + 1 + h(v) \ln 2), \text{ for all } v \in \mathcal{F},$$

$$\sum_{t=\blacktriangleleft(v)}^{\blacktriangleright(v)} f_t(\mathbf{z}_{t,h(v)}) \le \sum_{t=\blacktriangleleft(v)}^{\blacktriangleright(v)} f_t(\mathbf{z}_{t,h(v)-1}) + \Xi, \text{ for all } v \in \mathcal{R} \setminus \mathcal{F}.$$

The rest analysis is identical to Theorem 2.

Finally, we focus on $\alpha$-exp-concave functions. The analysis is also similar. To bound the meta-regret, we can utilize Lemma 6 to obtain

$$\sum_{t=\blacktriangleleft(v)}^{\blacktriangleright(v)} f_t(\mathbf{z}_{t,h(v)}) \le \min \left\{ \sum_{t=\blacktriangleleft(v)}^{\blacktriangleright(v)} f_t(\mathbf{x}_{t,h(v)}), \sum_{t=\blacktriangleleft(v)}^{\blacktriangleright(v)} f_t(\mathbf{z}_{t,h(v)-1}) \right\} + \Xi',$$

where $\Xi' = 2\Gamma G D (2 + \frac{1}{\sqrt{\ln 2}}) + \frac{\Gamma^2}{2\alpha \ln 2}$, and $\Gamma$ is defined in (5). According to the theoretical guarantee of Maler (Wang et al., 2019, Theorem 1), we have

$$\sum_{t=\blacktriangleleft(v)}^{\blacktriangleright(v)} f_t(\mathbf{x}_{t,h(v)}) \le \sum_{t=\blacktriangleleft(v)}^{\blacktriangleright(v)} f_t(\boldsymbol{\epsilon}_t) + \left( 10GD + \frac{9}{2\beta} \right) (\gamma' + 10dh(v) \ln 2) \tag{25}$$

where $\beta = \frac{1}{2}\min\{\frac{1}{4GD},\alpha\}$. Combining Lemma 6 and (25), we have

$$\sum_{t=\blacktriangleleft(v)}^{\blacktriangleright(v)} f_t(\mathbf{z}_{t,h(v)}) \leq \sum_{t=\blacktriangleleft(v)}^{\blacktriangleright(v)} f_t(\boldsymbol{\epsilon}_t) + \Xi + \left(10GD + \frac{9}{2\beta}\right)(\gamma' + 10dh(v)\ln 2), \text{ for all } v \in \mathcal{F},$$

$$\sum_{t=\blacktriangleleft(v)}^{\blacktriangleright(v)} f_t(\mathbf{z}_{t,h(v)}) \leq \sum_{t=\blacktriangleleft(v)}^{\blacktriangleright(v)} f_t(\mathbf{z}_{t,h(v)-1}) + \Xi', \text{ for all } v \in \mathcal{R} \setminus \mathcal{F}.$$

The rest analysis is identical to Theorem 2.

