# OpenReview forum: "Logarithmic Switching Regret for Online Convex Optimization"
_ICML.cc/2026/Conference — ICML 2026 regular_

### Official Review · Reviewer_24d6 · 2026-02-15

**Soundness:** 3
**Presentation:** 3
**Significance:** 2
**Originality:** 3
**Overall Recommendation:** 5
**Confidence:** 2

**Summary:**

This paper studies the switching regret for strongly convex and exp-concave functions. The proposed algorithm, IRESET, is based on the prior development of RESET algorithm by Pasteris et al. (2024), which achieves the optimal switching regret for general convex functions. Naively applying the RESET algorithm to strongly convex and exp-concave functions yields switching regret bounds that are suboptimal by a factor of $O(\log T)$. To overcome this, the authors use Adapt-ML-Prod as the meta algorithm, and develop a new technique for transforming segment-tree-based bounds to the switching regret bound. The resulting bound reduces the $O(\log T)$ factor to $O(\log |\mathcal{I}|)$, where $|\mathcal{I}| \le T$ is the segment length. This bound is optimal up to an $O(\log |\mathcal{I}|)$ factor. Furthermore, the authors develop a universal algorithm for bounding the switching regret, based on the existing universal algorithm, Maler (Wang et al. 2019), for the static regret.

**Compliance With Llm Reviewing Policy:**

Affirmed.

**Final Justification:**

While my expertise on the existing literature, such as RESET, is limited, I believe this paper makes substantial contributions, and I have no major remaining concerns. Therefore, I maintain my original rating of accept.

**Key Questions For Authors:**

Overall, I am positive about this paper. I have no major points requiring clarification. The following questions are out of pure interest:

1. Can the algorithm be extended to bandit convex optimization?
2. Also, can it be extended to yield dynamic regret bounds? For example, comparators may move slightly within each segment, but jump across segments. Then, can we obtain switching regret bounds that depend on the path length in each segment?

**Limitations:**

yes

**Strengths And Weaknesses:**

S1. The paper presents a solid contribution to logarithmic bounds on the switching regret for strongly convex and exp-concave functions.

S2. While the proposed framework relies on the existing RESET algorithm, the authors introduce novel techniques for challenges unique to the strongly convex and exp-concave settings.

S3. The high-level idea of the algorithm, including the overview of the prior RESET algorithm, is clearly presented.

W1. The resulting bounds are still suboptimal by a factor of $O(\log |\mathcal{I}|)$. The authors are aware of this, and I'm not very negative about this. However, this makes the contribution less significant compared to the original RESET algorithm.

W2. Most technical details are deferred to the appendix. I understand the paper is already long and that some arguments are standard in the literature, but this makes it difficult to fully assess the technical contributions.

Minor comment:
It seems $\mathcal{F}_k$ is defined twice in Proposition 2.

---

> ### Author Rebuttal · Authors · 2026-03-31
>
> **Many thanks for the constructive reviews! We will revise our paper accordingly.**
>
> ---
>
> **Q1:** Most technical details are deferred to the appendix.
>
> **A1:** We appreciate the feedback regarding technical depth. In the revised version, we will relocate critical technical derivations and proof sketches from the appendix to the main body. This adjustment will allow for a more thorough evaluation of our key contributions while maintaining a clear presentation.
>
> ---
>
> **Q2:** It seems $\mathcal{F}_k$ is defined twice in Proposition 2.
>
> **A2:** We thank the reviewer for pointing this out. We will remove the redundant descriptive sentence and provide a single, consistent definition for $\mathcal{F}_k$ in the revised version to avoid any confusion.
>
> ---
>
> **Q3:** Can the algorithm be extended to bandit convex optimization?
>
> **A3:** We thank the reviewer for this insightful question regarding the extension of our framework to the bandit setting. In bandit convex optimization (BCO), the learner only has access to the function values. Based on the number of function values queried, BCO can be classified into one-point and two-point. We analyze the feasibility of this extension from two perspectives: the expert-algorithm level and the meta-algorithm level.
>
> * **Expert-algorithm:** For the one-point feedback model, to the best of our knowledge, there currently exist no algorithms that achieve optimal static regret for strongly convex or exp-concave functions. For the two-point feedback model, algorithms for exp-concave functions are similarly unavailable, while for strongly convex functions, existing methods (Agarwal et al., 2010) can achieve $O(d^2\log T)$. The lack of suitable expert-algorithms in most of these settings poses a fundamental barrier to instantiating our framework.
> * **Meta-algorithm:** The more critical challenge lies in the meta-algorithm. The Adapt-ML-Prod algorithm employed in our framework relies on linearized losses constructed from gradient information, which is unavailable in the bandit setting. Extending such second-order aggregation methods to BCO remains an open problem, and it is currently unclear how to address this challenge.
>
> **References**
>
> Agarwal, A., Dekel, O., and Xiao, L. Optimal algorithms for online convex optimization with multi-point bandit feedback. COLT, 2010.
>
> ---
>
> **Q4:** Can it be extended to yield dynamic regret bounds? For example, comparators may move slightly within each segment, but jump across segments.Then, can we obtain switching regret bounds that depend on the path length in each segment?
>
> **A4:** We thank the reviewer for this excellent question. The prior work of Pasteris et al. (2024) has indeed provided optimal dynamic regret $O(\sqrt{(1+P_T)T})$ for general convex functions. However, their analysis crucially relies on using OGD as the expert-algorithm and exploiting $O(\sqrt{T}(1+P_T))$ dynamic regret bound. We remark that extending this to strongly convex or exp-concave functions presents fundamental difficulties, even when comparators move only slightly within each segment. The key challenge is that  OGD/ONS algorithms for strongly convex or exp-concave functions do not admit dynamic regret bounds that depend on the path length of the comparator sequence. This prevents the per-segment expert-regret from being bounded in terms of the path length, making it difficult to accommodate changing comparators. We consider this an important open problem and will discuss it in the revised version.

---

> > ### Author Rebuttal · Reviewer_24d6 · 2026-04-01
> >
> > I appreciate the authors' response. I have no further questions. I retain my score and wait for the discussion.

---

> > > ### Author Response · Authors · 2026-04-06
> > >
> > > Dear Reviewer 24d6,
> > >
> > > We sincerely thank you for your valuable comments and positive feedback. We will carefully revise our paper accordingly.
> > >
> > > Best
> > >
> > > Authors

---

### Official Review · Reviewer_7eLJ · 2026-02-17

**Soundness:** 3
**Presentation:** 2
**Significance:** 3
**Originality:** 2
**Overall Recommendation:** 4
**Confidence:** 3

**Summary:**

The authors study the problem of online convex optimization where the loss functions can be either strongly convex or exp-concave. In such a setting, they provide the first algorithm capable of attaining $O(\sum_{I}\log^2(I))$ regret. Finally, the author provide a universal algorithm which may handle convex, strongly convex and exp-concave functions simultaneously.

**Compliance With Llm Reviewing Policy:**

Affirmed.

**Final Justification:**

The rebuttal did not particularly change my understanding of the paper. I will keep my original (positive) score.

**Key Questions For Authors:**

See weaknesses. In particular, can you elaborate on the second point?

**Limitations:**

yes

**Strengths And Weaknesses:**

**Strengths**

Overall, the results provided in the paper are interesting. Indeed, the regret bound provided improve the state-of-the-art ones.


**Weaknesses**

My concerns on the paper are the following:

1. The algorithmic novelty is limited. The algorithm provided in the paper seems a combination of existing techniques. While I still believe that the combination is non trivial, this is indeed a weak point of the paper.
2. In Theorem 2, the $\Gamma$ factor is logarithmic in $T$. It is in contrast on the claim that the logarithmic factor are removed by the algorithm proposed in the paper.

Minor: the presentation can be improved. For instance, in the regret definition, $\boldsymbol{w}$ is used in place of $\boldsymbol{x}$ .

---

> ### Author Rebuttal · Authors · 2026-03-30
>
> **Many thanks for the constructive reviews! We will revise our paper accordingly.**
>
> ---
>
> **Q1:** The algorithmic novelty is limited. The algorithm provided in the paper seems a combination of existing techniques. While I still believe that thecombination is non trivial, this is indeed a weak point of the paper.
>
> **A1:** We would like to take this chance to emphasize the technical novelty of our work. First, we acknowledge that the design of IRESET builds on existing algorithms, including the meta-algorithm and the expert-algorithm. The same is also true for RESET (Pasteris et al., 2024), which employs Hedge as the meta-algorithm and OGD as the expert-algorithm.
>
> Although existing techniques are sufficient for the algorithm design, the main technical challenge in achieving switching regret guarantees lies in choosing an appropriate meta-algorithm and introducing a novel theoretical analysis based on segment tree structure. Specifically, for exp-concave/strongly convex functions, most existing meta-algorithms, such as FLH (Hazan and Seshadhri, 2007) or SCB (Jun et al., 2017), are inadequate for our problem. These methods incur $O(\log T)$ or $O(\sqrt{\vert I\vert})$ factor for the interval, which exist a large gap from our established lower bound. Therefore, we choose a meta-algorithm with a second-order bound, and prove that it delivers $O(1)$ meta-regret for exp-concave or strongly convex functions.
>
> More critically, the analysis of Pasteris et al. (2024) does not extend to these function classes considered in our work. Their recursion fundamentally relies on $\sqrt{\vert I\vert}$-type terms and collapses when the regret scales as $\log \vert I\vert$. Our work introduces a new analysis based on segment tree structure and a conversion from tree-based switching regret to a formulation in terms of the interval length. This is essential for eliminating the $\log T$ factor and for achieving bounds that nearly match the lower bounds in Theorem 1.
>
> ---
>
> **Q2:** In Theorem 2, the factor $\Gamma$ is logarithmic in $T$. It is in contrast on the claim that the logarithmic factor are removed by the algorithm proposed in the paper.
>
> **A2:** We thank the reviewer for this observation and apologize for the confusion. We clarify that the factor $\Gamma$ in Theorem 2 is *doubly logarithmic* in $T$, i.e., $\Gamma=O(\log\log T)$, and our regret bound does indeed suffer from an additional term $\log\log T$ due to the design of the meta-algorithm. Following prior work (Chernov & Vovk, 2010; Luo & Schapire, 2015), such double-logarithmic factors can be treated as constants in most cases. To make this clearer, in the revised version, we will explicitly state that our results remove the $O(\log T)$ factor present in prior strongly adaptive regret bounds, while a residual $O(\log\log T)$ dependence on $T$ remains. We will clarify this term more precisely in the results section to avoid any potential confusion.
>
> ---
>
> **Q3:** The presentation can be improved. For instance, in the regret definition, $w$ is used in place of $x$.
>
> **A3:** We thank the reviewer for this suggestion. We will improve the presentation and fix the notational inconsistency in the revised version, replacing $\mathbf{w}$ with $\mathbf{x}$ in the regret definition to ensure consistency with the notation used throughout the paper.

---

> > ### Author Rebuttal · Reviewer_7eLJ · 2026-04-01
> >
> > I would like to thank the Author for the responses. The answers did not particularly changed my evaluation and my understanding of the work. I will keep my current positive evaluation.

---

> > > ### Author Response · Authors · 2026-04-06
> > >
> > > Dear Reviewer 7eLJ,
> > >
> > > Many thanks for your insightful questions. We will revise our paper to clarify the points you raised and ensure the presentation is clearer. We also truly appreciate your positive evaluation of our work.
> > >
> > > Best
> > >
> > > Authors

---

### Official Review · Reviewer_zLgg · 2026-03-11

**Soundness:** 3
**Presentation:** 2
**Significance:** 2
**Originality:** 2
**Overall Recommendation:** 4
**Confidence:** 3

**Summary:**

This paper addresses a gap in Online Convex Optimization (OCO) for non-stationary environments by establishing nearly-optimal switching regret bounds for α-exp-concave and λ-strongly convex functions, which previously suffered from a suboptimal O(log T) penalty. The authors propose IRESET, a novel meta-algorithm that maintains multiple expert algorithms over Dense Geometric Covering intervals and sequentially aggregates their decisions using a second-order bounded meta-algorithm with linearized losses. To prove its efficacy, the authors develop a new theoretical framework utilizing recursive equations over a segment tree structure, successfully removing the dependence on the total time horizon T. This achieves nearly-optimal regret bounds of O(Σ_{I∈S} (1/λ) log^2 |I|) for λ-strongly convex functions, and O(Σ_{I∈S} (d/α) log^2 |I|) for α-exp-concave functions. Furthermore, by incorporating an existing universal expert (Maler), they introduce IRESET-Maler, a universal strategy that simultaneously achieves optimal or nearly-optimal switching regret across general convex, α-exp-concave, and λ-strongly convex functions without requiring any prior knowledge of the specific function types or their parameters.

**Compliance With Llm Reviewing Policy:**

Affirmed.

**Final Justification:**

After going over the rebuttal and other reviews, I have decided to increase my original rating from weak reject to weak accept. I think the paper has sufficient merit to be accepted to the conference. Still, I would not oppose its rejection if the AC decides in that direction.

**Key Questions For Authors:**

In addition to the issues raised in the weaknesses, some specific questions are listed below:

1.	Given that IRESET (Algorithm 1) requires instantiating an expert for every Dense Geometric Covering (DGC) interval that starts at time t and maintaining an ever-growing active set while sequentially aggregating them pairwise via Adapt-ML-Prod, what are the precise time and memory complexities of IRESET and IRESET-Maler at round t? Have you conducted any preliminary empirical simulations to observe the actual computational overhead?
2.	You transparently note that the O(∑_{I∈S} log^2∣I∣) bounds exhibit an O(log∣I∣) gap from the theoretical minimax lower limits established in Theorem 1. Is this gap an inherent algorithmic limitation of sequentially aggregating decisions via Adapt-ML-Prod over the segment tree, or do you suspect it is an artifact of the current analysis (e.g., bounds arising from Proposition 2)?
3.	In Equation 3, IRESET explicitly requires dividing by the product of the gradient bound (G) and domain diameter (D) to normalize the linearized losses for the Adapt-ML-Prod meta-algorithm. In purely online settings, these global constants are rarely known exactly a priori. How sensitive is the theoretical guarantee to mis-specifications or loose upper bounds of G and D, and could standard techniques, such as the "doubling trick," be integrated to dynamically guess these bounds without breaking the delicate segment tree analysis?
4.	Theorem 2 establishes that the switching regret bound for α-exp-concave functions scales linearly with the dimensionality d (O(∑_{I∈S} d/α log^2∣I∣)). In modern, high-dimensional machine learning tasks, this linear dependence could be prohibitive. Are there any structural assumptions, or alternative expert algorithms (other than ONS) that could be plugged into IRESET to alleviate this strict linear scaling with d?

**Limitations:**

While the authors honestly address one major theoretical limitation, they do not adequately discuss the practical limitations.

1.	Acknowledge Practical and Computational Limitations: The authors transparently acknowledge the theoretical limitation of their work, specifically, the remaining O(log∣I∣) gap from the theoretical minimax lower bound. However, they should also explicitly discuss the practical limitations of the IRESET algorithm. Maintaining and sequentially aggregating an ever-growing set of active experts for every Dense Geometric Covering interval introduces significant computational and memory overhead. Acknowledging this intractability for real-world software applications is crucial.
2.	Discuss Dimensionality Constraints: The authors should note the limitation regarding exponentially concave functions, where the switching regret bound scales linearly with the dimensionality d. They should discuss how this linear dependence impacts the algorithm's utility in modern, high-dimensional machine learning tasks.
3.	Address the Lack of Empirical Validation: The paper is purely theoretical. The authors should explicitly state that the numerical stability, actual runtime, and real-world performance advantages over existing baselines remain unverified in practice.

**Strengths And Weaknesses:**

Strengths:
•	Clear target problem + meaningful direction: Extending optimal switching-regret ideas beyond general convex losses to strongly convex and exp-concave losses is a natural and relevant next step.
•	Universal extension is a nice add-on: Using a universal static-regret method (Maler) inside the IRESET framework is a reasonable attempt to reduce parameter tuning.

Weaknesses:
•	Incrementality vs. Pasteris et al. (2024): I believe the difference from Pasteris et al. (2024) is (a) swapping Hedge for Adapt-ML-Prod and (b) reworking the recursion to handle log terms. As written, the contribution reads as a fairly direct extension and makes this paper only incremental. If the difference is significant, it should be articulated better. Also, the fact that the rate in the main result is suboptimal decreases the significance.
•	Presentation issues in Sec. 3.1:
o	Abstract is very long
o	The algorithm description in Sec 3.1. is dense and the difference to RESET is not very clear.
o	The pseudocode ordering is confusing: Alg. 1 step (20) says “observes the loss function f_t(\cdot)”. Isn’t f_t used before in the updates between lines 14-18?
o	The beginning of Sec. 3.3 is largely a mini related-work survey on universal online learning, inserted near the end. This interrupts the flow; it would be stronger either earlier or tightened to only what is strictly needed for Thm. 3.
•	While the paper proposes a novel meta-algorithm (IRESET) and a universal strategy (IRESET-Maler), it is entirely theoretical. It contains absolutely no experimental results, empirical evaluations, or simulations. Because the authors claim their method resolves issues with prior algorithms and provides a universal strategy, the complete absence of practical evaluation makes it impossible to verify the algorithm's real-world runtime, numerical stability, or actual performance advantages over existing baselines.
•	The primary motivation of the paper is to achieve optimal switching regret for strongly convex and exp-concave functions. However, the authors explicitly acknowledge in their "Clarification" section that their derived bounds, O(log^2∣I∣), still exhibit a gap of O(log∣I∣) from the theoretically established minimax lower bounds. Therefore, the paper only achieves "nearly-optimal" bounds, leaving the core question of strict optimality unresolved.
•	In Algorithm 1 (IRESET), the algorithm must dynamically create and maintain an active set of experts for every Dense Geometric Covering (DGC) interval that starts at any given round t. At each round, it sequentially combines the decisions of all active experts, which requires computing weights and intermediate outputs iteratively. While the authors note that the segment tree itself is only for theoretical analysis, actually running multiple instances of Online Newton Step (ONS) or Online Gradient Descent (OGD) concurrently, and aggregating them pairwise at every time step, is likely to introduce massive computational complexity and memory costs in practical software implementation.
•	The switching regret bound achieved for α-exp-concave functions explicitly scales linearly with the dimensionality d of the problem, resulting in a bound of O(∑_{I∈S} d/α log^2∣I∣). In modern machine learning tasks where the parameter dimension d can be extremely high (e.g., thousands or millions of parameters), this linear dependence could render the theoretical guarantees practically meaningless, making the algorithm scale poorly.
•	To ensure the meta-algorithm (Adapt-ML-Prod) functions correctly, IRESET requires the losses to be strictly normalized between . To do this, Equation 3 explicitly requires the algorithm to divide by the product of the gradient bound (G) and the domain diameter (D). This means the user must know the exact maximum gradient norm and domain bounds a priori to run the algorithm. In many pure online learning settings, these global constants are unknown beforehand, and guessing them incorrectly could either break the theoretical bounds or lead to overly conservative updates.

---

> ### Author Rebuttal · Authors · 2026-03-30
>
> **Many thanks for the constructive reviews! We will revise our paper accordingly.**
>
> ---
>
> **Q1:** I believe the difference ... makes this paper only incremental.
>
> **A1:** We would like to take this chance to emphasize the technical novelty of our work. First, we acknowledge that the design of IRESET builds on existing algorithms, including the meta-algorithm and the expert-algorithm. The same is also true for RESET (Pasteris et al., 2024), which employs Hedge as the meta-algorithm and OGD as the expert-algorithm.
>
> Although existing techniques are sufficient for the algorithm design, the main technical challenge in achieving switching regret guarantees lies in choosing an appropriate meta-algorithm and introducing a novel theoretical analysis based on segment tree structure. Specifically, for exp-concave/strongly convex functions, most existing meta-algorithms, such as FLH (Hazan and Seshadhri, 2007) or SCB (Jun et al., 2017), are inadequate for our problem. These methods incur $O(\log T)$ or $O(\sqrt{\vert I\vert})$ factor for the interval, which exist a large gap from our established lower bound. Therefore, we choose a meta-algorithm with a second-order bound, and prove that it delivers $O(1)$ meta-regret for exp-concave or strongly convex functions.
>
> More critically, the analysis of Pasteris et al. (2024) does not extend to these function classes considered in our work. Their recursion fundamentally relies on $\sqrt{\vert I\vert}$-type terms and collapses when the regret scales as $\log \vert I\vert$. Our work introduces a new analysis based on segment tree structure and a conversion from tree-based switching regret to a formulation in terms of the interval length. This is essential for eliminating the $\log T$ factor and for achieving bounds that nearly match the lower bounds in Theorem 1.
>
> ---
>
> **Q2:** Experimental results.
>
> **A2:** We have conducted additional experiments: https://anonymous.4open.science/r/Switching_experiment-D630/Switching_experiment.pdf. We choose the ijcnn1 dataset from LIBSVM Data, and consider the online classification problem. To simulate the changing environment, the labels of samples are flipped every $200$ rounds.
>
> ---
>
> **Q3:** Computational complexity ...?
>
> **A3:** Assuming the per-iteration complexity of each expert-algorithm is $O(1)$, the computational complexity of IRESET and IRESET-Maler are $O(\log t)$ and $O(\log^2 t)$ at round $t$, respectively. The memory complexity follows the same order.
>
> We would like to emphasize that these complexities are **consistent with all existing strongly adaptive and universal algorithms** in the literature, and do not incur any additional computational overhead. This cost is a necessary price for achieving adaptivity and universality. Regarding empirical evaluation, we conduct a comparing the computational overhead of different algorithms below.
>
> **Table 1:** Running times (s) of different methods for three types of convex functions.
>
> |      Loss       | OGD  |  SCB  | IRESET-OGD/ONS | IRESET-Maler |
> | :-------------: | :--: | :---: | :------------: | :----------: |
> | Strongly convex | 2.53 | 11.95 |     24.58      |    27.25     |
> |   Exp-concave   | 4.64 | 32.71 |     63.22      |    69.26     |
> | General convex  | 2.14 | 30.67 |     55.40      |    62.89     |
>
> ---
>
> **Q4:** ... Is this gap an inherent algorithmic limitation ...?
>
> **A4:** We believe the gap is primarily an artifact of the segment tree analysis. Specifically, the gap arises from Proposition 2: for a segment of length $\vert\mathcal{I}\vert$, the number of fundamental vertices can be up to $O(\log\vert\mathcal{I}\vert)$. In the analysis of Pasteris et al. (2024) for general convex functions, this does not cause an issue because the regret term scales as $\sqrt{2^k}$ for a fundamental vertex at height $k$, and summing it over fundamental vertices yields $\sum_{k\in[N]}\sqrt{2^k}\in O(\sqrt{2^N})$, which matches the optimal rate. However, in our setting, the regret term scales as $\ln(2^k)$, and summing it over fundamental vertices results in $\sum_{k\in[N]}\ln(2^k)\in O(\ln^2(2^N))$ instead of the optimal $O(\ln (\vert\mathcal{I}\vert))$. This is the fundamental reason why our analysis incurs the additional factor.
>
> ---
>
> **Q5:** IRESET explicitly requires ...?
>
> **A5:** We clarify that bounded gradients ($G$) and bounded domain diameter ($D$) are standard assumptions in the OCO literature, adopted by most existing online learning studies. As for the doubling trick, it requires restarting the algorithm upon parameter updates, which would disrupt the DGC interval structure.
>
> ---
>
> **Q6:** Are there .. alternative (other than ONS) ...?
>
> **A6:** We clarify that the linear dependence on $d$ is an inherent characteristic of the exp-concave setting. The best static regret bound for exp-concave functions is $O(\frac{d}{\alpha}\log T)$, achieved by ONS, which is already known to be minimax optimal (Ordentlich & Cover, 1998).

---

> > ### Author Rebuttal · Reviewer_zLgg · 2026-04-03
> >
> > I appreciate the rebuttal effort, acknowledge the additional clarifications and experiments of the authors. While helpful, I still believe that my previous rating still reflects my recommendation of the work.

---

### Official Review · Reviewer_QouF · 2026-03-12

**Soundness:** 3
**Presentation:** 3
**Significance:** 2
**Originality:** 2
**Overall Recommendation:** 4
**Confidence:** 4

**Summary:**

The paper studies OCO under the "switching regret" metric, which is the sum of static regret of arbitrary segmentations of the horizon. Building on the RESET algorithm for general convex losses, it proposes I RESET, which replaces the meta-layer with a second-order aggregation scheme (AdapMLProd), which helps removing the usual extra $\log T$ overhead that appears in strongly adaptive methods.

With OGD or ONS as base experts, the paper achieves switching regret bounds of order $\sum_{I\in S}\log^2 |I|$ for strongly convex and exp-concave losses, and it further claims a ``universal" variant based on Maler that simultaneously handles (detects) general convex, strongly convex, and exp-concave losses. The paper is clearly motivated by, and based on,  the prior result of Pasteris et al.\ on optimal switching regret for general convex OCO and by Maler-style universal OCO.

**Compliance With Llm Reviewing Policy:**

Affirmed.

**Final Justification:**

Some of the main concerns have been resolved.

**Key Questions For Authors:**

1. Given the tension outlined in the Significance section, can you identify and motivate a specific, practical regime where replacing $\log T$ with $\log |\mathcal{I}|$ provides a qualitatively stronger guarantee?

2. You acknowledge the $O(\log |\mathcal{I}|)$ gap between your upper bounds and the lower bounds from Theorem 1. Do you believe the lower bound in Theorem $1$ might be simply too loose because it relies on the strong assumption of known segmentation? I suspect the $\log(|I|)$ might not be avoidable in a lower bound construction that does not assume the learner knows the segmentation. If this is the case, this would strengthen your results because they will be tight then.

**Limitations:**

The authors are clear about the limitation of their work, as evident by the clarification section.

**Strengths And Weaknesses:**

Overall, the paper appears technically sound. The main ideas are reasonable, the proof strategy is coherent. There are some typos, and in few places the notation makes exposition harder than necessary to verify the details, e.g., the filled vs non-filled triangles, the height of a node. etc. More specifically, (i) there is likely missing factor of $2$ in the normalization of the linearized losses since the main text and the appendix differ; (ii) there is notation inconsistency in the regret definition, where $w$ appears to be used instead of $x$; (iii) there is a missing root near equation $(10)$? the $4\varGamma GD$ factor.

The presentation in the main paper is coherent. The high-level motivation is clear, the narrative is easy to follow, and the contribution is positioned well enough for a reader familiar with online learning and adaptive regret. The main paper does a good job of communicating the big picture: the goal is to improve the meta-layer so that one can avoid the extra $\log T$ factor inherited from prior approaches. On the other hand, clarity and attention in the appendix seems to drop, e.g., (i) Several equations are dense and would benefit from more verbal explanation., e.g., the use of Adapt-ML-Prod results feels somewhat abrupt.  (ii) the appendix should include a short review of the relevant guarantee being instantiated, the necessary assumptions, and to explain how the quantities in this paper map to that guarantee.


Re significance, the problem is relevant, and the paper makes a contribution by removing the global time horizon dependency from the local segment bounds. By avoiding sleeping experts (and probably FLH), the authors improve the bound factors from $O(\log |\mathcal{I}| \log T)$ to $O(\log^2 |\mathcal{I}|)$, offering a resolution to the question of strongly convex and exp-concave function gurantees under switching regret. Nonetheless, the significance of replacing $\log T$ by $\log |\mathcal I|$ is, in my view, limited by the tradeoff between interval length and the number of segments. To see this, consider a segmentation into intervals of comparable length $L := |\mathcal I|$, so that the number of segments is on the order of $T/L$.

(i) To obtain \emph{more than a constant-factor} improvement over prior bounds that scale with $\log T$, one needs $\log L = o(\log T),$ i.e., the interval length must be \emph{subpolynomial} in $T$. In particular, if $L = T^\alpha$ for any fixed $\alpha \in (0,1)$, then $\log L = \alpha \log T,$  so the claimed improvement is a multiplicative constant factor.

(ii) On the other hand, when $L$ is very small, the number of segments becomes correspondingly large. In the extreme case $L = O(1)$, the segmentation contains $\Theta(T)$ intervals. More generally, with intervals of length $L$, the total bound scales as
 \begin{equation}
   \frac{T}{L} \mathrm{polylog}(L),
  \end{equation}
    which is sublinear only when $L$ itself grows with $T$ (which puts us back in case $1$).
\end{itemize}

In short, to obtain a sublinear overall switching-regret bound, the interval length must grow with $T$; however, once $L$ grows polynomially, the replacement of $\log T$ by $\log L$ yields only a constant-factor gain. Thus, while the improvement is mathematically clean, the paper does not clearly isolate a regime in which it yields a qualitatively stronger guarantee.

The paper does not introduce an entirely new paradigm, but it does make a meaningful new combination of ideas: it adapts the RESET + Adapt-ML-Prod (+ Maler to adapt function families). That is a reasonable theoretical contribution. In particular, the originality lies less in inventing a brand-new primitive and more in identifying the right meta-algorithmic replacement and showing that it integrates with the interval construction in a useful way.

---

> ### Author Rebuttal · Authors · 2026-03-30
>
> **Many thanks for the constructive reviews! We will revise our paper accordingly.**
>
> ---
>
> **Q1:** There are some typos, and in few places the notation makes exposition harder than necessary to verify the details. Specifically, ...
>
> **A1:** We sincerely thank the reviewer for the careful examination of our work. We address each point below:
>
> * **Missing factor of $2$ in the normalization:** We confirm that Equation (3) contains a typo, and the denominator should be $2GD$ instead of $GD$. We will correct this in the revised version.
> * **Regret definition:** We will replace $\mathbf{w}$ with $\mathbf{x}$ in the regret definition to ensure consistency with the notation used throughout the paper.
> * **Missing root $4\Gamma GD$ near Equation (10):** We clarify that this term is not missing but has been absorbed into the preceding term. Specifically, it is incorporated into $2\Gamma GD(2+1/\sqrt{\ln 2})$. We will make this clearer in the revised version.
>
> ---
>
> **Q2:** Given the tension outlined in the Significance section, can you identify and motivate a specific, practical regime where replacing $\log T$ with $O(\log \vert\mathcal{I}\vert)$ provides a qualitatively stronger guarantee?
>
> **A2:** We thank the reviewer for this thoughtful question. As the reviewer correctly points out, our improvement reduces to a constant factor when segment lengths are uniform and polynomial in $T$. We acknowledge this observation and would like to offer the following complementary perspectives.
>
> First, in many practical non-stationary environments, segment lengths are inherently **heterogeneous** rather than uniform. Our bound $O(\sum_{\mathcal{I}\in \mathcal{S}}\log^2 \vert\mathcal{I}\vert)$ adapts to each individual segment length, whereas the prior bound $O(\sum_{\mathcal{I}\in \mathcal{S}}\log \vert\mathcal{I}\vert\log T)$ penalizes every segment by the global horizon $T$, including very short ones. This per-segment adaptivity is beneficial whenever the segmentation contains a mixture of long and short intervals.
>
> Second, we identify a concrete regime where the improvement is qualitatively stronger: when segment lengths are polylogarithmic in $T$, i.e., $\vert\mathcal{I}\vert=\log^c (T)$ for some constant $c>0$. Consider, for example, online classification in streaming sensor networks where environmental conditions (e.g., temperature, humidity, lighting) shift the data distribution periodically. As the deployment horizon $T$ grows (e.g., months or years of continuous operation), the duration of each stationary period does not grow proportionally, sensor environments may shift every few hundred or thousand readings regardless of total deployment length, yielding segment lengths that scale as polylog$(T)$. In this regime:
>
> - The number of segments is $\Theta (T/\log^c(T))$,
> - Our bound yields $O((T/\log^c (T)\cdot (\log\log T)^2)$,
> - The prior bound yields $O((T/\log^c (T)\cdot (\log\log T)\log T)$.
>
> The ratio between the two bounds is $\Theta (\log T/\log\log T)$, which grows unboundedly with $T$. This constitutes a qualitatively stronger guarantee, not merely a constant-factor improvement.
>
> ---
>
> **Q3:** You acknowledge the $O(\log \vert\mathcal{I}\vert)$ gap between your upper bounds and the lower bounds from Theorem 1. Do you believe the lower bound in Theorem might be simply too loose because it relies on the strong assumption of known segmentation? I suspect the $O(\log \vert I\vert)$ might not be avoidable in a lower bound construction that does not assume the learner knows the segmentation. If this is the case, this would strengthen your results because they will be tight then.
>
> **A3:** We thank the reviewer for this insightful suggestion. If a matching lower bound could be established for the unknown segmentation setting, our results would indeed become tight.
>
> However, we would like to point out that proving lower bounds in the unknown segmentation setting is a difficult open problem. The unknown segmentation scenario is closely related to **strongly adaptive regret**, where the algorithm must perform well over all possible intervals simultaneously. To the best of our knowledge, no existing work has been able to establish minimax lower bounds for strongly adaptive regret, and all known results rely on static regret lower bounds to argue near-optimality, similar to our work.
>
> ---
>
> **We hope that our responses can address your concerns, and we would greatly appreciate it if you could re-evaluate the contributions of our work.**

---

> > ### Author Rebuttal · Reviewer_QouF · 2026-04-04
> >
> > Thank you for the detailed explanations, much appreciated.

---

> > > ### Author Response · Authors · 2026-04-06
> > >
> > > Dear Reviewer QouF,
> > >
> > > We are sincerely grateful for your valuable suggestions and for the score increase. In the revised version, we will provide further clarification regarding the significance of our results and the lower bound analysis in the context of unknown segments. Thank you once again for your feedback, which has substantially enhanced the quality of our work.
> > >
> > > Best
> > >
> > > Authors

---

### Decision · Program_Chairs · 2026-04-30

**Decision:**

Accept (regular)

**Comment:**

This paper studies switching regret in online convex optimization for strongly convex and exp-concave losses, extending prior work that obtained optimal guarantees only for general convex functions. The proposed IRESET framework combines interval-based experts with a second-order meta-algorithm and a new segment-tree-based analysis, leading to improved bounds that remove the dependence on the global horizon
$T$ from the per-segment guarantees. The paper also includes a universal variant covering multiple function classes. Overall, the reviewers found the paper technically sound and relevant.

The main concerns were that the algorithmic ingredients build on prior work and that the final bounds still have an extra
$O(\log|I|)$ gap relative to the lower bound. These are valid limitations, but I do not think they outweigh the contribution. The authors’ rebuttal clarified the nontrivial aspects of the analysis, explained the source of the remaining gap, and addressed several presentation issues. On balance, I view this as a solid theoretical contribution that should be of interest to the online learning community, and I recommend acceptance. I encourage the authors to include the rebuttal contents in the final version of the paper.